# Engineered factor Xa variants retain procoagulant activity independent of direct factor Xa inhibitors

Daniël Verhoef [1], Koen M. Visscher[2], C. Ruben Vosmeer[2], Ka Lei Cheung[1], Pieter H. Reitsma[1], Daan P. Geerke [2] & Mettine H.A. Bos [1]

The absence of an adequate reversal strategy to prevent and stop potential life-threatening bleeding complications is a major drawback to the clinical use of the direct oral inhibitors of blood coagulation factor Xa. Here we show that specific modifications of the substrate-binding aromatic S4 subpocket within the factor Xa active site disrupt high-affinity engagement of the direct factor Xa inhibitors. These modifications either entail amino-acid substitution of S4 subsite residues Tyr99 and/or Phe174 (chymotrypsinogen numbering), or extension of the 99-loop that borders the S4 subsite. The latter modifications led to the engineering of a factor Xa variant that is able to support coagulation in human plasma spiked with (supra-)physiological concentrations of direct factor Xa inhibitors. As such, this factor Xa variant has the potential to be employed to bypass the direct factor Xa inhibitor-mediated anticoagulation in patients that require restoration of blood coagulation.

[1] Division of Thrombosis and Hemostasis, Einthoven Laboratory for Vascular and Regenerative Medicine, Leiden University Medical Center, Albinusdreef 2, 2333 ZA Leiden, The Netherlands. [2] AIMMS Division of Molecular Toxicology, Department of Chemistry and Pharmaceutical Sciences, Vrije Universiteit, De Boelelaan 1108, 1081 HZ Amsterdam, The Netherlands. Correspondence and requests for materials should be addressed to M.H.A.B. (email: M.H.A.Bos@lumc.nl)

The human hemostatic system protects against thrombosis and bleeding by balancing pro- and anticoagulant stimuli through an intricate network of enzymatic reactions governed by (pro)enzymes, (pro)cofactors, and inhibitors, collectively known as the coagulation cascade. Blood coagulation factor X (FX) plays a pivotal role in this system as it, once activated and assembled into the prothrombinase complex, converts prothrombin to thrombin. Thrombin is the key regulatory enzyme of the coagulation cascade and, among others, converts soluble fibrinogen to insoluble fibrin strands, which serve to stabilize the platelet-based primary blood clot. The spatiotemporal assembly of the prothrombinase complex is tightly regulated and occurs exclusively on negatively charged membrane surfaces (of activated cells or platelets), where activated factor X (FXa) assembles with its cofactor activated factor V (FVa) in the presence of calcium ions[1]. This process is initiated through the activation of FX by the extrinsic (tissue factor (TF)-factor VIIa (FVIIa)-mediated) or intrinsic (factor VIIIa (FVIIIa)-factor IXa (FIXa)-mediated) pathways of coagulation. Once activated, FXa also propagates coagulation by activating other factors[2], including plasma FV in a phospholipid-dependent manner[3]. The interaction of FXa with its cofactor FVa is essential as it results in physiologically relevant catalytic rates of prothrombin activation[1, 4].

The (chymo)trypsin-like serine protease FXa circulates in plasma as a 59 kDa zymogen glycoprotein and consists of two chains that are covalently linked by a disulfide bond. The N-terminal light chain contains a vitamin K-dependent gamma-carboxyglutamic acid-rich (GLA) domain and two epidermal growth factor-like (EGF) domains; the C-terminal heavy chain consists of an activation peptide and a serine protease domain. The FXa serine protease domain adopts the classical two β-barrel fold of chymotrypsin-like serine proteases, with the catalytic triad residues His57, Asp102, and Ser195 (chymotrypsinogen numbering) situated in the active site cleft that is located between the two β-barrels[5]. While the catalytic triad in conjunction with the oxyanion hole residues regulate substrate cleavage, the active site subpockets S1 and S4 control substrate recognition and binding. In the S1 subsite, this interaction is facilitated through a salt bridge between Asp189 and a positively charged side chain/moiety from the substrate/ligand. The aromatic S4 subpocket, which is formed by residues Tyr99, Phe174, and Trp215, contributes via hydrophobic interactions. The macromolecular substrate specificity and affinity are primarily directed through exosite binding[6], which involves surface regions in the serine protease domain that are distinct from the active site[7]. Proteolytic removal of the FX activation peptide induces maturation of the serine protease domain through conformational rearrangements, resulting in proper alignment that allows for engagement of the exosite and active site regions[8, 9]. Apart from substrate binding, the mature active site also readily interacts with the naturally occurring inhibitors of coagulation. Tissue factor pathway inhibitor (TFPI) tightly binds both the TF–FVIIa–FXa complex as well as free FXa[10]. The principal inhibitor of freely circulating FXa is the irreversible serine protease inhibitor antithrombin (AT)[11].

Active site inhibition of procoagulant serine proteases including FXa has been the focus of anticoagulant drug discovery for over a decade[12]. This has led to the clinical approval of several orally active, synthetic inhibitors of FXa for the prophylactic management of stroke in atrial fibrillation and prevention and treatment of venous thrombosis. These so-called direct oral anticoagulants (DOACs) currently include the direct FXa inhibitors rivaroxaban[13], apixaban[14], and edoxaban[15]. By reversibly engaging the active site of FXa with high affinity, the small molecules effectively block the catalytic activity of both free and prothrombinase-assembled FXa. However, a major drawback to their use is the absence of an adequate reversal strategy to prevent and stop potential life-threatening bleeding complications associated with anticoagulant therapy. Here we present human FXa variants that display a reduced sensitivity to inhibition by the direct FXa inhibitors due to modifications in the active site region, which are based on exceptional structural adaptations found in FX variants that are expressed in the venom of specific Elapid snakes. Using a combined computational and biochemistry approach, we have uncovered the mechanistic basis of the

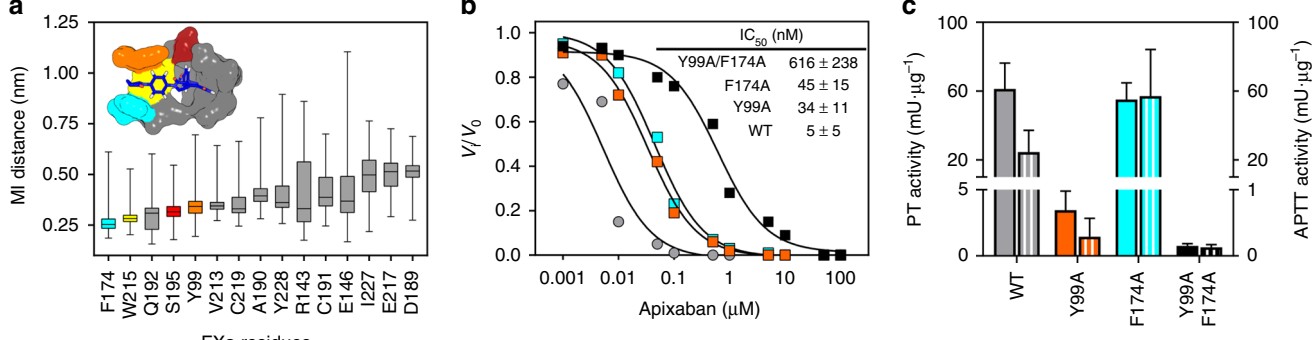

**Fig. 1** The S4 subsite of factor Xa coordinates apixaban binding and inhibition. **a** The minimal interatomic (MI) distances from the side chains of FXa residues to apixaban were calculated every nanosecond during a 750 ns MD simulation of apixaban-bound human FXa. *Box plots* (with *whiskers* from minimum to maximum) display the distribution of the observed MI distances for each of the 15 FXa residues with the shortest side chain to apixaban MI distance (sorted by average). The surface representation (*inset*) depicts the FXa-bound apixaban (*blue*) configuration throughout the MD simulation. Color coding (corresponding to *box plots*): S4 subsite residues Tyr99 (*orange*), Phe174 (*cyan*), and Trp215 (*yellow*), catalytic residue Ser195 (*red*), and others (*gray*). **b** The rate of peptidyl substrate conversion by 5 nM of RVV-X-activated wild-type (*gray circles*, WT), Tyr99Ala (*orange squares*, Y99A), Phe174Ala (*cyan squares*, F174A), or double mutant (*black squares*, Y99A/F174A) FX was determined with saturating amounts of the cofactor FV-810 (30 nM) and anionic phospholipid vesicles (PCPS, 50 μM) in the absence ($V_O$) or presence ($V_i$) of increasing apixaban concentrations (0.001–100 μM). The *lines* were drawn following nonlinear regression analysis of the data sets, and the fitted parameters for IC$_{50}$ ± 1 standard deviation of the induced fit are shown in the *inset*. The data are the means of two independent experiments. **c** The specific extrinsic (PT Activity; *filled bars*) or intrinsic (APTT Activity; *striped bars*) clotting activity of wild-type (WT), Tyr99Ala (Y99A), Phe174Ala (F174A), or double mutant (Y99A F174A) FX from conditioned media was determined as described in 'Methods' by dividing the PT or APTT clotting activity over the FX antigen concentration. The data represent the average ± 1 standard deviation of three representative high-producing stable cell lines per FX variant

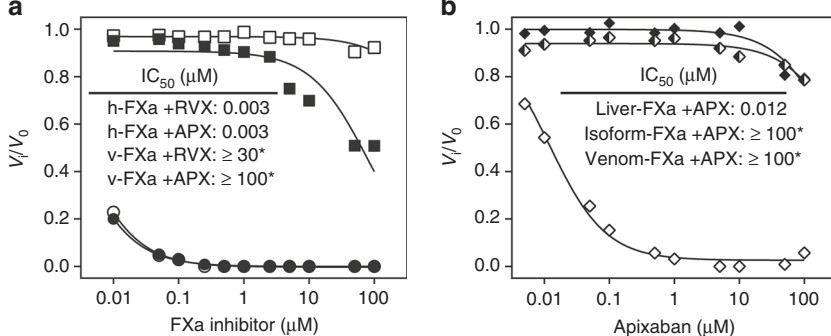

**Fig. 2** Functional characterization of the effect of the direct factor Xa inhibitors on factor Xa paralogs. **a** The rate of peptidyl substrate conversion by purified recombinant *P. textilis* venom FXa (10 nM; *squares*) or plasma-derived human FXa (2 nM; *circles*) was determined in the absence ($V_O$) or presence ($V_i$) of increasing concentrations (0.01–100 μM) of apixaban (APX; *open symbols*) or rivaroxaban (RVX; *closed symbols*). **b** The rate of peptidyl substrate conversion by RVV-X activated *P. textilis* venom FX (*closed squares*), isoform FX (*semi-closed diamonds*), or liver FX (*open diamonds*) was determined in the absence ($V_O$) or presence ($V_i$) of increasing apixaban concentrations (0.01–100 μM). **a**, **b** The lines were drawn following nonlinear regression analysis of the data sets, and the fitted parameters for IC$_{50}$ are shown in the *inset*. The data are the means of two independent experiments. *For these experiments, FXa inhibition was inefficient, precluding an accurate assessment of the IC$_{50}$ values

FXa inhibitor-sensitivity of these variants and demonstrate their effectiveness as potential bypassing agents in plasma containing direct FXa inhibitors.

## Results

**Inhibitor-resistance via disruption of S4 subsite binding.** Comparison of crystal structures of human FXa in complex with the direct FXa inhibitors apixaban (PDB 2P16) or rivaroxaban (PDB 2W26) revealed highly similar ligand-binding configurations, as both inhibitors occupy the FXa substrate binding S1 and S4 subsites through interactions with an almost identical set of amino acids[13, 14]. The X-ray structures further revealed that occupation of the S4 subsite is mediated, in part, by nonpolar stacking interactions, in which the P4 ring of the inhibitors is sandwiched between the aromatic side chains of Tyr99 and Phe174 (chymotrypsinogen numbering). To assess the molecular requirements for direct FXa inhibitor binding in more detail, we performed 750 ns molecular dynamics (MD) simulations of the FXa–apixaban complex (starting from the 2P16 crystal structure). During MD, apixaban adopted a stable position in the active site cleft of FXa, as reflected by the low minimal interaction distances (≤0.5 nm) between apixaban and many side chains of the residues lining the binding pocket, such as Phe174, Trp215, Gln192, Ser195, and Tyr99 (Fig. 1a). The narrow distribution of close-range contacts between apixaban and the side chain atoms of the S4 pocket residues Tyr99 and Phe174 confirmed stabilization of apixaban in the S4 subsite.

The contribution of S4 residues Tyr99 and Phe174 to apixaban binding was assessed in vitro by generating human FXa variants comprising either Tyr99Ala, Phe174Ala, or both mutations. While the single amino-acid substitutions resulted in a moderate reduction in apixaban inhibition (±8-fold enhanced IC$_{50}$), introducing both mutations increased the IC$_{50}$ of FXa inhibition by apixaban more than 100-fold (Fig. 1b). These results indicate that modifying the S4 subsite destabilizes binding of apixaban into the FXa active site. In addition, upon mutating Tyr99 to Alanine, the FX-specific clotting activity was drastically reduced (5% and 1% residual PT and APTT activity, respectively) and was essentially lost in combination with the Phe174Ala replacement (Fig. 1c). These findings suggest that although both Tyr99 and Phe174 contribute to apixaban binding, Tyr99 is of key importance to both the active site conformation and function of FXa. This is consistent with the fact that Tyr99 is not only evolutionary conserved in a wide range of FX species

(Supplementary Fig. 1), but also essential to active site maturation in the homologous blood coagulation serine protease FIXa[16, 17]. Alternatively, modifications targeting other amino acids that coordinate apixaban binding (Fig. 1a) may potentially disrupt docking of apixaban into the active site while preserving clotting activity.

Sequence analysis of FX species further revealed that FX paralogs found in various Elapid snakes comprise a heterologous insertion directly N-terminal to Tyr99 (between Thr95 and Lys96; Supplementary Fig. 1)[18]. These insertions result in an extended 99-loop (His91-Asp102) that borders the S4 subsite. Surprisingly, characterization of recombinantly prepared FXa paralogs from the Elapid snake *Pseudonaja textilis* showed that both the venom and isoform proteases, unlike any FXa species known to date, are highly insensitive to direct FXa inhibitors. This was demonstrated by a minimal 10,000-fold increase in the IC$_{50}$ of FXa inhibition by apixaban or rivaroxaban (Fig. 2a, b). While venom FXa is uniquely expressed in the venom gland[19], isoform FXa is both expressed in the liver and venom gland and represents an intermediate between *P. textilis* venom and liver FXa[20, 21]. In contrast, *P. textilis* liver FXa, which comprises a shorter 99-loop (Supplementary Fig. 1), is efficiently inhibited by submicromolar concentrations of apixaban (Fig. 2b). These results suggest that an extended 99-loop mediates a reduced sensitivity towards direct FXa inhibitors.

To elucidate the molecular mechanism that is at the basis of the apixaban resistance in *P. textilis* isoform FXa, we performed MD simulations using the crystal structure of isoform FXa (PDB 4BXW)[22]. Initial docking of apixaban showed that isoform FXa is able to accommodate apixaban in the active site in an orientation similar to that observed for human FXa (Supplementary Fig. 2). This is consistent with the observation that the conformation of the S4 subsite is conserved in both FXa variants (Supplementary Fig. 2). In isoform FXa, the 99-loop adopts an elongated and helical conformation (Supplementary Fig. 2) with high structural flexibility. The latter is demonstrated by the relatively high B-factors of the 99-loop obtained from 750 ns MD simulations of apixaban-bound and unbound isoform FXa (Supplementary Fig. 3). Five MD simulations of the isoform FXa–apixaban complex were independently performed, each was initiated from an identical conformation but with different atomic starting velocities. During these 750 ns MD simulations, we observed displacement of the isoform FXa 99-loop, as reflected by root-mean-square deviation (RMSD) values of ≥0.4 nm (Supplementary Fig. 3a–e). In addition, the following events were observed:

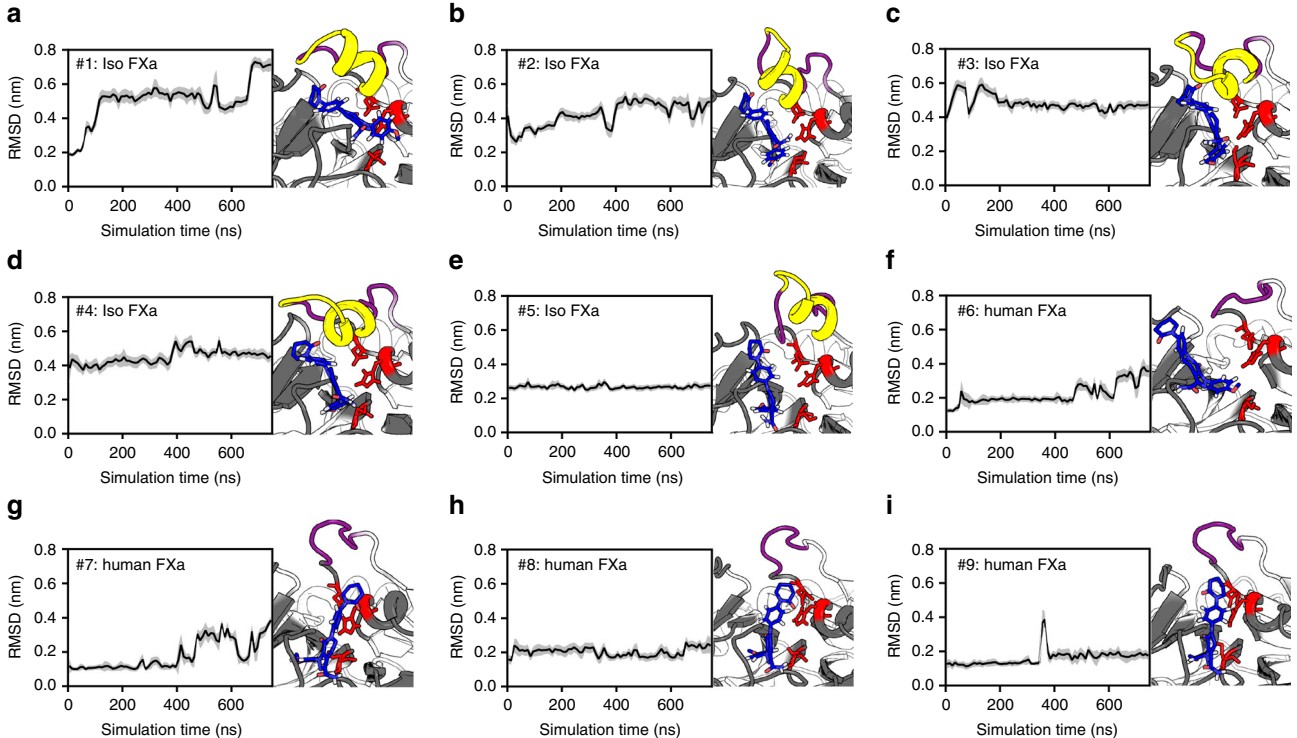

**Fig. 3** Molecular dynamics (MD) simulations of apixaban bound to factor Xa variants. Root-mean-square deviations (RMSDs) of atomic positions of the apixaban ligand during independent 750 ns MD simulations of apixaban binding to either *P. textilis* isoform FXa (Iso FXa, five independent MD simulations (**a**–**e**)) or to human FXa (four independent MD simulations (**f**–**i**)) are presented as block averages over 10 ns intervals. The corresponding single standard deviation interval is indicated (*gray* density). For each independent simulation, the molecular configurations at 750 ns are depicted, in which apixaban (*green*), the 99-loop (*magenta*), the isoform FXa extended 99-loop region PQKAYKFDL (*yellow*), and the catalytic triad residues (*red*) are highlighted. Molecular configurations at 250 and 500 ns of each simulation are compared with those at 750 ns in Supplementary Fig. 4

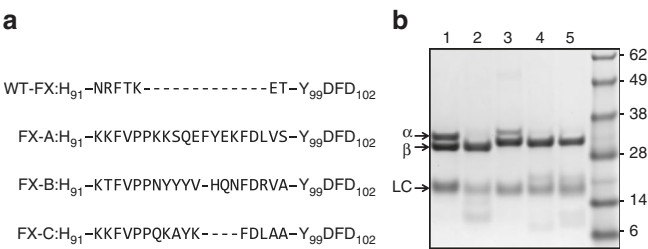

**Fig. 4** Snake–human factor X(a) variants. **a** An alignment of region His91-Asp102 in human FX (WT-FX) and FX variants A (FX-A), B (FX-B), and C (FX-C) is shown. The sequences inserted between His91 and Tyr99 originate from *P. textilis* venom FX (FX-A), from *T. carinatus* venom FX (FX-B), or from *P. textilis* isoform FX (FX-C). **b** Proteins (3 μg per lane) were subjected to SDS-PAGE under reducing conditions and visualized by staining with Coomassie Brilliant Blue. Lane 1: plasma-derived FXa; lane 2: recombinant wild-type FXa; lane 3: FXa variant A; lane 4: FXa variant B; lane 5: FXa variant C. The protein bands corresponding to the heavy chain derived from α or β FXa (α/β), the FXa light chain (LC), and the apparent molecular weights (kDa) of the standards are indicated. While autoproteolytic excision of the C-terminal portion of FXa-α (residues 436–447) yields the β form of FXa, both isoforms are functionally similar with respect to prothrombinase assembly, prothrombin activation, antithrombin recognition, and peptidyl substrate conversion[62]. The purified products of wt-FXa and FXa variants B and C migrate predominantly as FXa-β; FXa variant A migrates as a 50/50 mixture of α and β FXa. The data are representative of two independent experiments

(a) apixaban displacement from the S1 subsite, combined with a repositioning within the S4 subsite relative to its starting configuration (Fig. 3a, Supplementary Fig. 4a), (b) apixaban displacement from its initial docked conformation, partial dissociation from the S1 subsite and repositioning within the S4 subsite (Fig. 3b–d, Supplementary Fig. 4b–d), or (c) a stable apixaban-binding conformation in the S1 and S4 subsites that mirrors the initial docking pose of apixaban (Fig. 3e, Supplementary Fig. 4e) and is similar to the apixaban configuration in the crystallized human FXa–apixaban complex (Supplementary Fig. 2). Interestingly, this anchoring of apixaban in the S1 site was accompanied by the lowest flexibility in the displaced 99-loop (Supplementary Fig. 3e). In the other four isoform FXa–apixaban simulations, we observed substantial movement of the 99-loop (Supplementary Fig. 3a–d), similar to the simulations of unbound isoform FXa (Supplementary Fig. 3f–j). This indicates significant mobility of the 99-loop in both the apixaban-bound and -unbound states. The displacement of the 99-loop and the rapid displacement of apixaban observed for most MD simulations are indicative of steric hindrance between apixaban and the structurally flexible isoform FXa 99-loop, impairing apixaban binding. This could explain why isoform FXa is practically insensitive toward direct FXa inhibitors. This is further supported by MD simulations of apixaban binding to human FXa that lacks the extended 99-loop typical to isoform FXa. During these simulations (Fig. 3f–i, Supplementary Fig. 4f–i), partial dissociation of apixaban from the S1 subsite was observed in a single simulation only (at 600 ns; Fig. 3f). Furthermore, apixaban repositioning from the S4 subsite occurred during another individual simulation, but in a reversible manner (at 400 ns; Fig. 3g). Other than these two displacement

**Table 1 Kinetic constants for macromolecular or peptidyl substrate cleavage and prothrombinase assembly**

| | Prothrombin[a] $K_m$ (µM) | Prothrombin[a] $k_{cat}$ (min$^{-1}$) | Cofactor Va[a] $K_{d, app}$ (nM) | S2765[a] $K_m$ (µM) | S2765[a] $k_{cat}$ (s$^{-1}$) | S2765[b] $K_m$ (µM) | S2765[b] $k_{cat}$ (s$^{-1}$) |
|---|---|---|---|---|---|---|---|
| pd-FXa | 0.31 ± 0.07 | 1880 ± 136 | 0.41 ± 0.05 | 59 ± 19 | 47 ± 4 | 29 ± 8 | 36 ± 3 |
| wt-FXa | 0.41 ± 0.08 | 1243 ± 89 | 1.44 ± 0.41 | 33 ± 7 | 27 ± 2 | 26 ± 8 | 21 ± 2 |
| FXa-A | 0.14 ± 0.04 | 122 ± 8 | 0.81 ± 0.26 | 39 ± 7 | 9 ± 1 | 243 ± 50 | 11 ± 1 |
| FXa-B | 0.25 ± 0.07 | 239 ± 20 | 0.85 ± 0.28 | 49 ± 8 | 10 ± 1 | 249 ± 29 | 15 ± 1 |
| FXa-C | 0.22 ± 0.04 | 370 ± 19 | 0.57 ± 0.06 | 60 ± 11 | 17 ± 1 | 216 ± 28 | 21 ± 1 |

The kinetic constants for the enzyme [a]prothrombinase or [b]FXa were obtained as described in 'Methods'. Fitted values ± 1 standard deviation of the induced fit are representative of two to three independent experiments

**Table 2 Kinetic parameters for the inhibition of factor Xa variants**

| | Antithrombin $k_{2, uncatalyzed}$ (M$^{-1}$ s$^{-1}$ × 10$^3$) | Antithrombin $k_{2, UFH}$ (M$^{-1}$ s$^{-1}$ × 10$^6$) | TFPIα $K_i$ (nM) | rTAP $K_i$ (nM) | Apixaban IC$_{50}$ (nM) | Edoxaban IC$_{50}$ (nM) |
|---|---|---|---|---|---|---|
| pd-FXa | 1.55 ± 0.13 | 3.59 ± 0.39 | ND | 0.87 ± 0.18 | 2 ± 0.2 | 3 ± 2 |
| wt-FXa | 4.07 ± 0.20 | 3.09 ± 0.64 | 1.62 ± 0.25 | 0.70 ± 0.04 | 1 ± 0.2 | 2 ± 1 |
| FXa-A | 0.12 ± 0.02 | 1.55 ± 0.46 | 4.35 ± 1.32 | 26.30 ± 8.45 | 93 ± 23 | 23 ± 14 |
| FXa-B | 0.41 ± 0.03 | 1.02 ± 0.40 | 4.41 ± 1.89 | 7.92 ± 1.88 | 652 ± 200 | 218 ± 31 |
| FXa-C | 0.95 ± 0.11 | 6.73 ± 3.61 | 6.46 ± 1.63 | 20.24 ± 6.56 | 716 ± 247 | 375 ± 215 |

ND not determined
Fitted values ± 1 standard deviation of the induced fit are representative of two to three independent experiments

events, apixaban maintained its original orientation in human FXa during the 750 ns MD simulations.

**Characterization of chimeric FXa.** We next aimed to investigate whether insertions within the 99-loop similar to those found in snake FX (Supplementary Fig. 1) reduce the sensitivity of human FXa toward direct FXa inhibitors. To this end, human–snake FX chimeras were constructed in which the human sequence His91-Tyr99 was replaced with the homologous region of venom (FX-A) or isoform (FX-C) *P. textilis* FX, or of *Tropidechus carinatus* venom FX (FX-B) (Fig. 4a). Following stable FX expression in HEK293 cells, purified chimeric FXa variants were subjected to SDS-PAGE analysis, which showed all FXa variants to migrate similar to plasma-derived (pd-FXa) and recombinant wild-type FXa (wt-FXa) (Fig. 4b).

Evaluation of the kinetics of prothrombin conversion in the presence of saturating amounts of the FVa-like cofactor FV-810[23] and anionic phospholipids revealed that the rate of prothrombin activation by the chimeric variants was 3- to 10-fold reduced (Table 1), which may be indicative of a modified active site conformation. Nonetheless, the FXa variants displayed an up to threefold enhanced affinity for prothrombin as compared to wt-FXa and an apparent affinity for the cofactor Va that was similar to that of human FXa (Table 1). This shows that all chimeric variants are able to efficiently assemble into the prothrombinase complex and engage with the macromolecular substrate prothrombin.

**Extension of the FXa 99-loop impairs active site binding.** Analysis of the direct FXa inhibitor-dependent inhibition of prothrombin activation revealed a dramatic increase in half-maximum inhibition for all FXa variants. Apixaban or edoxaban inhibition of FXa assembled into prothrombinase was least efficient for variants B and C, as we observed an up to ~700-fold increase in IC$_{50}$, while the inhibition of variant A was ~10- to 90-fold impaired as compared to human FXa (Table 2). These data indicate that, consistent with the MD simulations (Fig. 3a–e), insertion of the snake venom His91-Tyr99 regions results in impaired binding of the S4 subsite in human FXa. As the cofactor FVa may also affect active site binding by apixaban, we assayed

apixaban-inhibition of FXa-C in the presence and absence of FVa. Interestingly, we observed a twofold higher IC$_{50}$ in the presence of factor Va and anionic phospholipids (Supplementary Fig. 5). These findings indicate that prothrombinase-assembled FXa-C may be more resistant to inhibition by apixaban to some extent, which could result from structural constraints imposed by the interaction of FXa with FVa.

To further explore the effect of the insertions on the FXa active site, the binding and cleavage of the peptidyl substrate S2765 was examined. We observed an up to ninefold increase in $K_m$ and twofold reduced $k_{cat}$ for the uncomplexed FXa variants compared to wt-FXa (Table 1). This further confirms that the FXa variants have a slightly altered activity and a modified active site. We next assessed the second-order rate constants of FXa inhibition by the active site-directed inhibitor AT[17]. Inhibition of FXa variant A proceeded at a 33-fold reduced rate relative to wt-FXa, whereas that of variants B and C was 10- and 4-fold decreased, respectively (Table 2). The impaired AT inhibition was restored upon the addition of unfractionated heparin (UFH), resulting in similar rates of AT inhibition for all FXa species, consistent with previous reports[24]. This shows that, unlike the FXa active site, the heparin-binding exosite in FXa[25] is not affected by insertion of the snake venom His91–Tyr99 regions. In addition, we investigated the binding affinities ($K_i$) of the active site inhibitors TFPIα and recombinant tick saliva anticoagulant protein (rTAP)[26] toward the chimeric FXa variants. Essentially, binding constants for both TFPIα and rTAP were perturbed, as rTAP binding was at least 11-fold reduced while TFPIα inhibition was at most fourfold reduced in the chimeric FXa variants (Table 2). The difference between the binding constants of TFPIα and rTAP toward chimeric FXa corroborates structural data that report a more extensive buried surface area between TFPIα–FXa (>1700 Å$^2$)[27] and rTAP–FXa (~900 Å$^2$)[28]. Moreover, inhibition of FXa by rTAP involves S4 subsite occupation[28], while inhibition of FXa by the Kunitz-II domain of TFPIα entails reorganization of the active site cleft through Tyr99 side chain reorientation[27]. Collectively, the attenuated inhibition of the FXa chimeras by AT, TFPIα, and rTAP is in agreement with the notion that extension of the 99-loop impairs active site engagement.

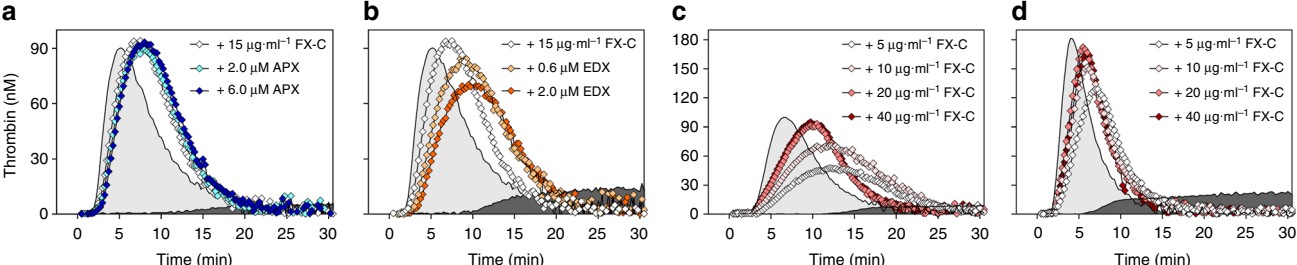

**Fig. 5** Zymogen factor X-C corrects thrombin generation in plasma spiked with apixaban. **a**, **b** Thrombin generation (TG) was measured for 30 min at 37 °C in FX-depleted plasma supplemented with 1 U ml$^{-1}$ FX-C (15 µg ml$^{-1}$) in the absence (*open symbols*) or presence of 2 or 6 µM apixaban (**a**, APX, *blue symbols*) or 0.6 or 2 µM edoxaban (**b**, EDX, *orange symbols*) with 2 pM tissue factor (TF) and 20 µM PCPS. Thrombin generation was initiated with CaCl$_2$ and a thrombin fluorogenic substrate as detailed in 'Methods'. The TG profiles of FX-depleted plasma supplemented with 1 U ml$^{-1}$ (7 µg ml$^{-1}$) wt-FX in the absence (*light gray* area under the curve) or presence (*dark gray* area under the curve) of 2 µM apixaban (**a**) or 0.6 µM edoxaban (**b**) are shown. **c**, **d** Thrombin generation was measured for 30 min at 37 °C in normal pooled plasma in the presence of 2 pM (**c**) or 6 pM (**d**) TF, supplemented with increasing concentrations FX-C (5–40 µg ml$^{-1}$), 2 µM apixaban, and 20 µM PCPS. The TG profiles of normal pooled plasma in the absence (*light gray* area under the curve) or presence (*dark gray* area under the curve) of 2 µM apixaban are shown. All curves are representatives of three to six independent experiments

Assessment of the catalytic efficiency of the FXa variants toward the macromolecular and peptidyl substrates suggests that this may be modulated by the length and/or composition of the region between His91 and Tyr99. When comparing all FXa derivatives, variant FXa-C that comprises the shortest His91–Tyr99 sequence (Fig. 4a) retained the highest $k_{cat}$ toward both substrates (Table 1). The same trend was observed for AT inhibition in the absence of UFH (Table 2). From this we conclude that all of the sequences inserted between His91 and Tyr99 negatively affect active site engagement. The relatively short amino-acid insertion in variant C is derived from isoform FXa in which it adopts a helical conformation (Supplementary Fig. 2)[22]. We speculate that this conformation may be maintained in FXa-C, resulting in a more structured 99-loop that compromises FXa catalytic activity least. Whether this results from a relatively limited extension over the active site and/or a different effect on overall protein motion and subpocket S4 flexibility between the FXa variants remains to be determined[29].

The relation between 99-loop architecture and substrate specificity has also been made clear in structural and biochemical studies on kallikreins. These trypsin-like serine proteases comprise 99-loops that vary greatly in length and can range from 2 to up to 22 additional residues between His91 and Tyr99, the latter concerns a kallikrein-like salivary toxin (BLTX) from the North American shrew[30, 31]. In general, variation of 99-loop length has significantly impacted enzymatic diversification in kallikreins. For example, the elongated 99-loop of human kallikrein-related peptidase-2 has been shown to extend over the active site and function as a regulator of enzyme activity through zinc binding[32]. Alternatively, the BLTX 99-loop has been implicated to contribute to an increased catalytic efficiency in silico[33]. The elongated 99-loops of the venom FXa paralogs may therefore be viewed in a broader perspective that highlights the general significance of 99-loop architecture in enzymatic diversification of trypsin-like serine proteases.

**Chimeric FX(a) as hemostatic bypassing agent.** The ability to activate prothrombin in the presence of physiological concentrations of direct FXa inhibitor potentially enables chimeric FXa to restore hemostasis in plasma inhibited by such anticoagulants. Functionality of the chimeric variants was therefore assessed by calibrated automated thrombography in plasma spiked with direct FXa inhibitors[34]. To do so, we first assessed FXa-initiated thrombin generation (TG) in FX-depleted plasma in the absence of FXa inhibitors. Thrombin generation initiated

by FXa variant C resulted in a thrombin peak height and endogenous thrombin potential (ETP) similar to wt-FXa-initiated TG, while these parameters were modestly reduced following initiation with variant A or B (Supplementary Table 1). As expected, addition of a physiological concentration of apixaban (2 µM) significantly impaired wt-FXa-initiated TG. In contrast, the TG parameters were not affected upon initiation with FXa variant C (Supplementary Table 1). As FXa-C displayed superior TG parameters, we consequently investigated whether the zymogen form of variant C (FX-C) would be able to restore the apixaban-dependent defect in TG. To assess this, thrombin generation was initiated with a limiting tissue factor concentration (TF, 2 pM) in FX-depleted plasma supplemented with plasma concentrations of zymogen FX. Other than a 1.4-fold delay in time to peak with FX-C present (Supplementary Table 2), comparable TG curves and parameters were obtained following supplementation with FX-C or wt-FX in the absence of inhibitor (Fig. 5a). Consistent with previous observations, FX-C-dependent TG was fully sustained both in the presence of physiological (2 µM) and supra-physiological (6 µM) concentrations of apixaban (Fig. 5a) or edoxaban (Fig. 5b), while the TG parameters were severely impaired following wt-FX supplementation (Supplementary Table 2).

The ability of FX-C to restore TG in the presence of apixaban was also examined in normal pooled plasma (NPP) from healthy individuals. Similar to previous findings, apixaban almost fully inhibited thrombin generation in NPP (Fig. 5c, d). Addition of increasing concentrations of zymogen FX-C to NPP restored TG both under conditions of a limited TF-trigger (Fig. 5c) and upon initiation with a high TF concentration (Fig. 5d). This was demonstrated by a normalization of the peak height in the presence of 20–40 µg ml$^{-1}$ FX-C, while the time to peak was 1.3- to 1.6-fold prolonged (Supplementary Table 3). Assessment of TG upon addition of 40 µg ml$^{-1}$ FX-C in the absence of apixaban indicated that no significant surplus of thrombin was generated as all TG parameters were within the range of those of NPP (Supplementary Table 3).

As the zymogen form of variant C typically displayed a TF-dependent delay in time to peak thrombin formed, we investigated whether activation of FV by variant FXa was perturbed. The FXa-dependent activation of FV is essential for the early phase of TG, thereby mediating the subsequent burst of thrombin formed[35]. While proteolysis of full-length plasma-derived FV by FXa-C did result in generation of the FVa heavy chain and light chain activation products, FXa-C appeared to

cleave FV at one or more sites that are distinct from cleavage by wt-FXa, indicated by a differential fragmentation profile (Supplementary Fig. 6). The altered substrate recognition of FXa-C was further assessed by cleavage analyses of factors IX, XI, and protein C (Supplementary Figs. 7–9), which suggests that FXa-C maintains selective substrate specificity. However, the potential acquisition of additional protein targets cannot be excluded. Furthermore, the delay in time to peak was not due to a defect in activation by the intrinsic tenase complex[36], as the kinetic parameters for FX-C activation by the intrinsic (FVIIIa/FIXa) tenase complex were unperturbed (Supplementary Fig. 10). Another explanation could lay in the notion that activation of FX-C by the extrinsic or intrinsic tenase results in a partly zymogen-like conformation of the FXa active site. Given that assembly of FXa into the prothrombinase complex at least partially corrects an impaired active site[8, 9, 37, 38], we examined the affinity of the chimeric FXa variants towards the peptidyl substrate S2765 in the presence of saturating amounts of the cofactor Va and anionic phospholipids. Indeed, prothrombinase complex assembly of the FXa variants almost fully corrected the defective peptidyl substrate binding while the $k_{cat}$ values were not significantly altered (Table 1). These results indicate that the delay in time to peak observed for chimeric FX-C in the TG assays reflects the additional time that is required by FX-C to engage FVa and become fully active. Importantly, studies on the protease–zymogen equilibrium of trypsin-like proteases have shown that the S4 subsite is involved in stabilization of the protease state[29, 39, 40]. As the 99-loop is allosterically linked to the S4 subsite, insertions within the 99-loop may indeed shift the protease equilibrium toward a more zymogen-like state in the chimeric FXa variants.

## Discussion

Through a combined computational and biochemical approach we have successfully pioneered the engineering of direct FXa inhibitor insensitivity in FX(a). Active site inhibition of FXa by direct FXa inhibitors requires binding of the ligand's P4 moiety into the S4 subsite through apolar stacking interactions with residues Tyr99 and Phe174. Our data have shown that disruption of apixaban binding is achieved by either replacing S4 subsite residues Tyr99 and/or Phe174 with Alanine, or by extension of the 99-loop that forms part of the S4 subsite. By replacement of the wild-type FXa 99-loop sequence with that of the P. textilis isoform FXa paralog, we obtained a FXa variant (FX(a)-C) that was highly resistant to the direct FXa inhibitors (apixaban IC$_{50}$: ~715 nM; edoxaban IC$_{50}$: ~375 nM). This variant was also able to restore thrombin generation in apixaban-spiked human plasma when added at a 2- to 4-fold plasma concentration (20–40 µg ml$^{-1}$). In addition, FX-C was capable of restoring thrombin generation at supra-physiological concentrations of apixaban or edoxaban. Finally, FX-C induced no in vitro hypercoagulability when present at a fourfold plasma concentration in the absence of FXa inhibitor. These properties therefore allow FX-C to function as a bypassing agent in human plasma in order to reverse direct FXa inhibitor anticoagulation.

Thus far, a specific and adequate reversal strategy for the treatment of serious and life-threatening bleeding complications associated with the FXa inhibitor-mediated anticoagulant therapy is not clinically available[41]. Consequently, semi-specific reversal strategies to overcome FXa inhibition have been developed[42–44], of which two are based on modified forms of FXa. Andexanet alfa is a GLA domainless, catalytically inactive FXa variant that functions as a decoy by trapping the FXa inhibitors through stoichiometric binding[42]. Another reversal approach takes advantage of a zymogen-like FXa variant (FXa-I16L) that

comprises an impaired active site, but displays full catalytic activity when assembled into prothrombinase[9]. The reversal strategies that are based on scavenging the inhibitory compounds from the circulation require a high dose of catalytically inactive FXa molecules (0.9 or 1.8 g protein per patient)[42] or synthetic cationic molecules (≤300 mg per individual)[43]. Our approach of bypassing the action of FXa-specific DOACs through diminished FXa inhibitor binding could have several benefits over the scavenging strategies. First, restoring hemostasis with inhibitor-insensitive FX would require administration of milligrams rather than grams of protein to equal or double the FX plasma concentration (10 µg ml$^{-1}$)[2]. In addition, a single administration of inhibitor-insensitive FX could potentially be sufficient to completely restore hemostasis, as the circulatory half-life of FX (34–40 h) exceeds that of the direct FXa inhibitors apixaban (~12 h) and edoxaban (10–14 h)[2, 45, 46]. Moreover, the ability of an inhibitor-insensitive form of FX to sustain thrombin generation in plasma irrespective of the FXa inhibitor concentration should facilitate ease of dosing in patients with limited background information.

In summary, chimeric FX-C could have the potential to serve as rescue therapeutic agent to overcome the effect of synthetic FXa inhibitors in case of potentially life-threatening bleeding events or emergency surgical interventions. Further studies into FX-C dosing, half-life, and mitigation of FXa inhibitor-dependent bleeding will therefore be required to assess its in vivo potential.

## Methods

**Reagents.** Rivaroxaban, apixaban, and edoxaban were obtained from Alsachim (Illkirch, France) or Adooq Bioscience (Irvine, CA, USA), weighted and dissolved in vehicle (10% (v/v) EtOH, 10% (v/v) Glycerol, 10% (v/v) PEG400 (Sigma Aldrich, St Louis, Mo, USA), and 70% (v/v) Dextrose (5% solution)), and aliquots were stored at −20 °C. The peptidyl substrate methoxycarbonylcyclohexylglycylglycyl-Arg-pNA (SpecXa) was obtained from Sekisui Diagnostics (Stamford, CT, USA), and the peptidyl substrates H-D-Phe-Pip-Arg-pNA (S2238), N-α-benzyloxycarbonyl-D-Arg-Gly-Arg-pNA (S2765), and pyroGlu-Pro-Arg-pNA (S2366) were obtained from Instrumentation Laboratories (Bedford, MA, USA). All tissue culture reagents were from Life Technologies (Carlsbad, CA, USA) except insulin-transferrin-sodium selenite (ITS), which was from Roche (Basel, Switzerland). Calibrator and fluorescent substrate (FluCa) were from Thrombinoscope (Maastricht, the Netherlands). FX-depleted human plasma, Neoplastine CI Plus 10 prothrombin time (PT) reagent, and Triniclot automated activated partial thromboplastin time (APTT) reagent were obtained from Diagnostica Stago (Paris, France). NPP was obtained from Sanquin (Amsterdam, the Netherlands). UFH was obtained from LEO Pharma (Ballerup, Denmark). Polybrene was obtained from Sigma Aldrich. All functional assays were performed in Hepes-buffered Saline (HBS: 20 mM Hepes, 0.15 M NaCl, pH 7.5) supplemented with 5 mM CaCl$_2$ and 0.1% (w/v) PEG8000 (assay buffer). The mammalian expression vector pCMV4 carrying recombinant human FX[47] or recombinant P. textilis venom FX, the pcDNA3.1 vector carrying Furin proprotein convertase, and a baby hamster kidney (BHK) cell line stably expressing a constitutively active partial B-domainless recombinant form of human FV (FV-810)[23] were generous gifts from Rodney Camire (Children's Hospital of Philadelphia). Human embryonic kidney (HEK) 293 cells were obtained from ATCC (CRL-1573; Manassas, VA, USA). Cell lines were monthly checked for mycoplasma contamination employing the Venor®GeM mycoplasma detection kit (Minerva Biolabs, Berlin, Germany), and mycoplasma contamination was eliminated using Plasmacure™ and Plasmocin™ as prescribed by the supplier (InvivoGen, San Diego, CA, USA). Small unilamellar phospholipid vesicles (PCPS) composed of 75% (w/w) hen egg L-phosphatidylcholine (PC) and 25% (w/w) porcine brain L-phosphatidylserine (PS) (Avanti Polar Lipids, Alabaster, AL, USA) were prepared by drying the lipid solution (24 mg PC, 8 mg PS) under a stream of nitrogen into a thin layer. Following resuspension of the lipids in 11 ml HBS, the lipid suspension was maintained in an ice bath and was continuously sonicated for 30 min using a direct probe with an output of 18 W under a constant stream of nitrogen. The vesicle dispersion was centrifuged at 35,000 rpm for 30 min at 24 °C in a Beckman ultracentrifuge (SW 55 Ti swinging bucket rotor) to remove particles and large multilamellar vesicles. Subsequently, the dispersion was centrifuged at 40,000 rpm for 3 h at 24 °C, and the clear supernatant (6 ml) was stored at 4 °C. The phospholipid concentration of the PCPS solution was determined using an organic phosphate determination[48].

**Proteins.** Plasma-derived human factors Xa (pd-FXa), IX (FIX), IXa, and XI (FXI), human prothrombin, protein C (PC), and antithrombin (AT), DAPA, and corn trypsin inhibitor (CTI) were from Haematologic Technologies (Essex Junction, VT,

USA). Human tissue factor (TF, Innovin) was obtained from Siemens (Newark, NY, USA), human plasma-derived factor VIII (FVIII, Aafact) was from Sanquin (Amsterdam, the Netherlands), and RVV-X activator was from Diagnostica Stago. Paired antibodies to determine factor X antigen were obtained from Cedarlane (CL20245K; Burlington, Canada)[49]. The Q5 site-directed mutagenesis kit and restriction endonucleases Bgl2, Hind3, and Apa1 were obtained from New England Biolabs (Ipswich, MA, USA). T4-DNA ligase was obtained from Roche. rTAP was a generous gift from Sriram Krishnaswamy (Children's Hospital of Philadelphia). Recombinant FV-810 was large-scale expressed in BHK cells in triple flasks (Nalge Nunc, Rochester, NY, USA) in Dulbecco's modified Eagle's medium/F-12 without phenol red supplemented with 2 mM L-glutamine, 100 U ml$^{-1}$ penicillin, 0.1 mg ml$^{-1}$ streptomycin, 0.25 µg ml$^{-1}$ amphotericin B, 100 µg ml$^{-1}$ geneticin, 10 µg ml$^{-1}$ ITS, and 2.5 mM CaCl$_2$. Conditioned media was collected for six consecutive days, centrifuged at 4000×$g$ to remove cellular debris, filtered over an 0.45 µm poly-ethersulfone membrane (Merck Millipore, Billerica, MS, USA), and supplemented with 1 mM benzamidine (Sigma Aldrich) prior to storage at −20 °C. Conditioned media (6 l) was thawed at 37 °C and applied at ambient temperatures to a 2.5 × 6 cm SP Sepharose Fast Flow column (GE Healthcare, Chicago, IL, USA) equili-brated in 20 mM Hepes, 0.15 M NaCl, 5 mM CaCl$_2$,1 mM benzamidine, pH 7.4. Following washing with the same buffer, bound protein was eluted with buffer containing 0.65 M NaCl. Fractions containing FV activity were stored at −80 °C. Following thawing at 37 °C, the fractions were pooled and diluted to 0.15 M NaCl in 20 mM Hepes, 5 mM CaCl$_2$, pH 7.4, stored on ice, and applied to a 10 × 100 mm POROS™ HQ 20 µm column (Applied Biosystems, Waltham, MS, USA). Following washing with the same buffer, bound protein was eluted with a linear 0.15–1 M NaCl gradient. Fractions containing FV activity were analyzed employing SDS-PAGE analysis, stored at −80 °C, pooled upon thawing at 37 °C, and ultrafiltrated employing Amicon Ultra-15 centrifugal filter units with 30 kDa molecular weight cutoff (Merck Millipore) to ~3 mg ml$^{-1}$ in 20 mM Hepes, 0.15 M NaCl, 5 mM CaCl$_2$, pH 7.4, and aliquots were stored at −80 °C. Molecular weights and extinction coefficients ($E_{0.1\%}$, 280 nm) of the various proteins used were taken as follows: prothrombin, 72,000 and 1.47; thrombin, 37,500 and 1.94; FX, 59,000 and 1.16; FXa, 46,000 and 1.16; FV-810, 216,000 and 1.54; FIX, 55,000 and 1.32; FIXa, 45,000 and 1.4; FXI, 160,000 and 1.34; PC, 62,000 and 1.45; AT, 58,000 and 0.62; and rTAP, 6900 and 2.56. For the FX(a) variants, all values for the human protein were used.

**Construction and expression of recombinant FX.** DNA constructs encoding FX variants comprising the Tyr99Ala, Phe174Ala, or both substitutions were prepared from the pCMV4 vector carrying wild-type human FX by site-directed mutagenesis and sequenced for consistency. DNA constructs encoding *P. textilis* isoform and liver FX paralogs and inserts coding for chimeric FX variants (FX-A, FX-B, and FX-C) were synthesized by Genscript (Piscataway, NJ, USA), subcloned into pCMV4-FX expression vector using either Bgl2 and Hind3 (FX paralogs) or Apa1 (chimeric FX) and T4-DNA ligase and sequenced for consistency. HEK293 cell lines stably expressing wild-type recombinant human FX, (chimeric) variants of FX, or *P. textilis* paralogs of FX were obtained following co-transfection of pCMV4-FX with pcDNA3.1-Furin vectors by Lipofectamine2000 according to the manufacturer's instructions. FX expression of transfectants was assessed by con-ditioning individual clones for 24 h in Dulbecco's modified Eagle's medium/F-12 without phenol red supplemented with 2 mM L-glutamine, 100 U ml$^{-1}$ penicillin, 0.1 mg ml$^{-1}$ streptomycin, 0.25 µg ml$^{-1}$ amphotericin B, 100 µg ml$^{-1}$ geneticin, 10 µg ml$^{-1}$ ITS, and 6 µg ml$^{-1}$ vitamin K (Konakion) (FX-specific expression media) and subsequently measuring the FX-specific PT/APTT clotting activity in a modified one-step assay by mixing conditioned media with FX-depleted human plasma in a 1:1 ratio. A reference curve of normal pooled plasma serially diluted in either assay buffer with 0.1% bovine serum albumin (BSA; for PT) or Owren–Koller diluent (for APTT), mixed in a 1:1 ratio with FX-depleted human plasma, was used to calculate the equivalent FX Units per ml plasma. Transfectants with the highest expression were expanded into a 6320 cm$^2$ cell factory (Thermo Scientific, Waltham, MA, USA) and conditioned for 24 h in FX-specific expression media. Conditioned media was collected for 10 consecutive days, centrifuged at 4000×$g$ to remove cellular debris, filtered over an 0.45 µm polyethersulfone membrane, and supplemented with 1 mM benzamidine prior to storage at −20 °C.

**Purification of FX(a).** Conditioned media (20 l) was thawed at 37 °C, applied to a size 6 A ultrafiltration hollow fiber cartridge using an Äkta flux 6 instrument (GE Healthcare), diafiltrated to ~500 ml in 20 mM Hepes, 0.15 M NaCl, pH 7.4, dia-lyzed to 20 mM Tris, 0.15 M NaCl, pH 7.4, and stored at −20 °C. Following thawing at 37 °C, the pool was applied at ambient temperatures to a 4.8 × 4 cm Q Sepharose Fast Flow column (GE Healthcare) equilibrated in 20 mM Tris, 0.15 M NaCl, pH 7.4. Following washing with the same buffer, bound protein was eluted with a linear 0.15–0.75 M NaCl gradient. Fractions containing FX activity were stored at −80 °C. Following thawing at 37 °C, the fractions were pooled and two times dialyzed at 4 °C for 2 h to 1 mM Na$_2$HPO$_4$/NaH$_2$PO$_4$, pH 6.8 (4 l), following overnight dialysis to 40 mM Na$_2$HPO$_4$/NaH$_2$PO$_4$, pH 6.8 (4 l). The dialysate was applied at ambient temperatures to a Bio-Scale CHT20-I hydroxyapatite column (Bio-Rad, Hercules, CA, USA) equilibrated in 40 mM Na$_2$HPO$_4$/NaH$_2$PO$_4$, pH 6.8[50]. Following washing with the same buffer, bound protein was first eluted with a linear 40–100 mM Na$_2$HPO$_4$/NaH$_2$PO$_4$ gradient, followed by a linear 100–400

mM Na$_2$HPO$_4$/NaH$_2$PO$_4$ gradient at a flow rate of 3.3 ml min$^{-1}$. Fractions con-taining FX activity were analyzed employing SDS-PAGE analysis, stored at −80 °C, pooled upon thawing at 37 °C, ultrafiltrated employing Amicon Ultra-15 centrifugal filter units with 30 kDa molecular weight cutoff to 5–10 mg ml$^{-1}$ in HBS, 50% (v/v) glycerol, and stored at −20 °C. The typical yield of fully γ-carboxylated recombinant FX was 0.9 mg l$^{-1}$ conditioned medium. Purified recombinant FX was activated with RVV-X (0.1 U mg$^{-1}$ FX), isolated by size-exclusion chromatography on a Sephacryl S200 HR column ($V_t$ 460 ml; GE Healthcare), and stored at −20 °C in HBS containing 50% (v/v) glycerol. Purified products were visualized by Coomassie Brilliant Blue staining employing SDS-PAGE analysis.

**Specific clotting activity.** The specific extrinsic clotting activity was determined using a modified FX-specific PT-based clotting assay. Purified FX samples were serially diluted to <170 nM in assay buffer with 0.1% BSA. In a typical assay, 25 µl of FX-depleted plasma was mixed with an equal volume of sample, followed by a 60 s incubation period at 37 °C. Coagulation was initiated after the addition of 50 µl PT reagent, and the coagulation time was monitored using a Start4 coagu-lation instrument (Diagnostica Stago). The specific intrinsic clotting activity was determined using a modified FX-specific APTT-based clotting assay. FX samples were serially diluted to <170 nM in Owren–Koller diluent. FX-depleted plasma (25 µl) was mixed with sample (25 µl) and APTT reagent (50 µl), followed by an 180 s incubation period at 37 °C. Coagulation was initiated after the addition of 50 µl of 25 mM CaCl$_2$, upon which the coagulation time was monitored. Reference curves consisted of serial dilutions of NPP.

**Macromolecular substrate activation.** Steady-state initial velocities of macro-molecular substrate cleavage were determined discontinuously at 25 °C in assay buffer[51]. Progress curves of prothrombin activation were obtained by incubating PCPS (50 µM), DAPA (10 µM), and prothrombin (1.4 µM) with human recom-binant FV-810 (20 nM) for 10 min, and the reaction was initiated by the addition of 0.1–1 nM pd-FXa, wt-FXa, FXa-A, FXa-B, or FXa-C. Samples (10 µl) were withdrawn at various time points (0–3 min) and quenched by mixing with 90 µl of 20 mM Hepes, 0.15 M NaCl, 50 mM EDTA, 0.1% (w/v) PEG8000, 2 µM wt-TAP, pH 7.5. Quenched samples were then further diluted in the same buffer lacking rTAP, and initial velocities of S2238 hydrolysis were determined in a SpectraMax M2e kinetic plate reader (Molecular Devices, Berks, UK). Measured rates were related to the concentration of thrombin from the linear dependence of initial velocity on known concentrations of thrombin determined in each experi-ment. The reported kinetic parameters and equilibrium binding constants for prothrombin ($k_{cat}$, $K_m$) and cofactor Va (FV-810; $K_{d, app}$) by pd-FXa and wt-FXa correspond to previous reported values[23]. Prothrombin conversion was assayed in the absence or presence of the direct FXa inhibitors apixaban or edoxaban (0.001–100 µM final) in order to determine IC$_{50}$ concentrations for each (recom-binant) FXa variant. Activation of FX variants by the intrinsic FVIIIa/FIXa tenase complex was achieved by incubating 40 nM FVIII with 100 nM of thrombin during 30 s, upon which 150 nM hirudin (Sigma Aldrich) was added. Progress curves of FX activation were obtained by incubating FIXa (0.5 nM), PCPS (20 µM), and FX (13–2200 nM) for 5 min at 25 °C, and the reaction was initiated by the addition of FVIIIa (5 nM). Samples (10 µl) were withdrawn at various time points (0–4 min) and quenched by mixing with 90 µl of 20 mM Hepes, 0.15 M NaCl, 50 mM EDTA, 0.1% (w/v) PEG8000, pH 7.5. Quenched samples were then further diluted in the same buffer, and initial velocities of SpecXa hydrolysis were determined in a SpectraMax M2e kinetic plate reader. Measured rates were related to the con-centration of wt-FXa or FXa-C from the linear dependence of initial velocity on known concentrations of wt-FXa or FXa determined in each experiment.

**Chromogenic substrate hydrolysis.** The kinetics of peptidyl substrate hydrolysis (SpecXa and S2765) were measured in assay buffer using increasing concentrations of substrate (10–800 µM) and initiated with free FXa (2–5 nM) or assembled into prothrombinase using the following conditions: PCPS (50 µM) and FV-810 (20 nM).

**Inhibition of FXa by antithrombin.** The rate of inactivation of FXa by AT was measured in assay buffer under pseudo-first-order rate conditions at ambient temperatures[52]. Uncatalyzed reactions were prepared in assay buffer containing 0.2–1.0 µM AT with 7.5 nM FXa. After 0–110 min, residual enzyme activity remaining as a function of time was determined after the addition of 250 µM SpecXa and monitoring the initial steady-state increase in absorbance at 405 nm. Catalyzed reactions were prepared in assay buffer containing 1–8 nM UFH, 0.5 µM AT, and were initiated with 5.0 nM FXa. After 0.5–5.0 min, residual enzyme activity remaining as a function of time was determined after the addition of 250 µM SpecXa supplemented with 1 mg ml$^{-1}$ polybrene. The rate of AT inhibition was determined by fitting the obtained values to an exponential decay function ($k_1$) and subsequent analysis by linear regression ($k_2$) using the GraphPad Prism software suite.

**Inhibition of FXa by TFPI and rTAP.** The overall dissociation constants ($K_i$) for TFPIα or rTAP binding to FXa were inferred from measurements of residual

enzyme amidolytic activity. For TFPIα, reactions were prepared in assay buffer containing 5 nM FXa and 2–40 nM TFPIα. After incubation for 3 h at ambient temperatures, residual enzyme activity was determined after the addition of 250 μM SpecXa and monitoring the initial steady-state increase in absorbance at 405 nm. For rTAP, reactions were prepared in assay buffer containing 1 nM FXa and 0.5–100 nM rTAP. After incubation for 30 min at ambient temperatures, residual enzyme activity was determined after the addition of 250 μM SpecXa and monitoring the initial steady-state increase in absorbance at 405 nm. The dissociation constants were determined by fitting the obtained values to the Morrison equation for tight binding using the GraphPad Prism software suite. The equation was constrained for the concentration of enzyme, substrate, and SpecXa Michaelis–Menten constant ($K_m$: pd-FXa, 138 μM; wt-FXa, 135 μM; FXa-A, 645 μM; FXa-B, 658 μM; FXa-C, 659 μM).

**Calibrated automated thrombography analysis**. Thrombin generation was adapted from protocols earlier described[34]. Thrombin generation curves were obtained by supplementing FX-depleted plasma with TF (2 pM final), CTI (70 μg ml$^{-1}$), PCPS (20 μM), and 1 U ml$^{-1}$ (prothrombin time FX clotting activity) of wt-FX (7 μg ml$^{-1}$) or chimeric FX-C (15 μg ml$^{-1}$) in the absence or presence of apixaban. Alternatively, thrombin generation curves were obtained by supplementing NPP with TF (2 or 6 pM final), CTI (70 μg ml$^{-1}$), PCPS (20 μM), and 10–40 μg ml$^{-1}$ of wt-FX or FX-C in the absence or presence of apixaban. Thrombin formation was initiated by adding substrate buffer (FluCa) to the plasma. FXa-initiated thrombin generation curves were obtained by supplementing FX-depleted plasma with CTI (70 μg ml$^{-1}$) and PCPS (20 μM). Thrombin formation was initiated by the addition of FXa premixed with apixaban in assay buffer without CaCl$_2$, and supplemented with a thrombin fluorogenic substrate in CaCl$_2$-containing buffer (FluCa). The final reaction volume was 120 μl, of which 80 μl was plasma. Thrombin formation was determined every 20 s for 30–60 min and corrected for the calibrator using Thrombinoscope software. The lag time, mean endogenous thrombin potential (ETP, the area under the thrombin generation curve), time to peak, peak thrombin generation, and velocity index were calculated from at least three individual experiments.

**MD simulations**. MD simulations were performed using GROMACS version 5.1.4[53], starting from the crystal structure with PDB 2P16[14] for apixaban in complex with wt-FXa, or from the structure obtained by docking apixaban in the substrate binding cleft of isoform FXa (using the 4BXW PDB structure of isoform FXa[22]). To prevent steric hindrance between apixaban and the side chain of Glu192, its chi-1 dihedral was adapted from −147.66° to 54.41° before docking, in order to open up the binding cleft and representing the orientation in human FXa. Docking was performed using the PLANTS software[54]. The center of the binding site was set to the corresponding center of geometry of apixaban in the wt-FXa structure with a search radius of 1.1 nm. The docking algorithm was ran under speed setting 1 and the resulting models were scored using CHEMPLP[54]. The third-ranked structure was selected based on binding pose similarity to wt-FXa in complex with apixaban.

During MD simulations, the protein was described using the GROMOS 54A7[55] force field and the system was numerically integrated with a time step of 2 fs, using a leap-frog integrator[56]. After initial energy minimization, the protein was solvated in a periodic dodecahedral box with a minimum distance of 1.4 nm between the solute and the box edges. The system was neutralized by simulating the system in a 0.15 M NaCl solution. After additional energy minimization, atomic velocities were randomly assigned according to Maxwell–Boltzmann statistics using different seeds for different simulations. The system was equilibrated in four subsequent 1 ns simulations at a constant number particles (N), volume (V), and temperature (T) (NVT), where the temperature was raised from an initial 100 to 200 K, and two simulations at a final temperature of 300 K. During the heating steps, protein heavy-atom positional restraining force constants were applied and gradually lowered (from 10,000 to 5000, 50, and 0 kJ mol$^{-1}$ nm$^{-2}$, respectively). After 5 ns unrestrained pre-equilibration, production simulations were run for 750 ns under NpT conditions in which the temperature was coupled weakly by velocity-rescaling[57] to an external bath at 300 K using a coupling constant of 0.1 ps. The pressure of the system was kept constant at a reference pressure of 1 bar, in an isotropic manner by coupling to a Berendsen barostat[58] using a coupling constant of 0.5 ps and a compressibility of $4.5 \times 10^{-5}$ bar$^{-1}$. All bond lengths were constrained during simulation using the LINCS algorithm[59] with a single iteration and a highest order of 4 in the constraint coupling matrix. Pairwise interactions were monitored using a Verlet neighbor list[60] with a buffer tolerance of 0.005 kJ mol$^{-1}$ ps$^{-1}$. Short-range electrostatic and van der Waals interactions up to a cutoff of 1.4 nm were evaluated every time step, and long-range electrostatic interactions were computed using a PME scheme[61] with cubic interpolation and a fourier grid spacing of 0.125 nm. Center of mass motion was removed every 100 steps.

Simulation data were written out every 50 ps for further analyses, and minimal atomic distances were monitored using the GROMACS analysis tool g_mindist, where minimal interatomic distances were calculated every 1 ns between residue side-chain and apixaban atoms. For each residue in the protein we calculated a distribution of these distances over the complete production run, and we subsequently sorted the residues by the distribution median. RMSDs during simulation of apixaban atomic positions were calculated after conformational fitting on the MD starting structure with respect to backbone atoms of the protein. Averaging over 10 ns intervals generated block-averaged time series for the analyzed data, and subsequently standard deviations were computed.

**Statistical analysis**. All in vitro data are presented as mean ± 1 standard deviation and are the result of at least three experiments, unless otherwise stated.

**Data availability**. All data supporting the findings of this study and code are available within the article and its Supplementary Information or from the corresponding author upon reasonable request.

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

## Acknowledgements

We thank Bouchra Eddounassi and Carlijn F. van der Sluijs for their technical assistance in establishing stable cell lines, preliminary protein characterization, and protein purification. We thank Laura F.H. Janssen, Kyra E. de Goede, and Suzan Spoelstra for their help in performing docking, Molecular Dynamics simulations, and data analyses. Financial support by The Netherlands Organization for Scientific Research (NWO, VIDI grant 723.012.105, D.P.G.) is gratefully acknowledged.

## Author contributions

D.V., K.M.V., C.R.V., D.P.G., and M.H.A.B. designed research; D.V., K.M.V., C.R.V., and K.L.C. performed research; D.V., K.L.C., P.H.R., D.P.G., and M.H.A.B. contributed new reagents/analytic tools; D.V., K.M.V., C.R.V., P.H.R., D.P.G., and M.H.A.B. analyzed data; and D.V., K.M.V., P.H.R., D.P.G., and M.H.A.B. wrote the manuscript.

## Additional information

**Competing interests:** D.V., P.H.R., and M.H.A.B. are co-inventors on a patent application related to this manuscript. P.H.R. owns equity in VarmX B.V. (a company founded to develop technology related to this manuscript). The remaining authors declare no competing financial interests.

