## [Peer Review File · Nature Communications]

Reviewers' Comments:

Reviewer #1 (Remarks to the Author):

Overview

The manuscript submitted by Dr. Bos and colleagues describes a novel approach to bypassing clotting factor Xa (FXa)-directed oral anticoagulants (DOACs), which is a high priority medical need. While one DOAC antidote has recently completed a stage 3b/4 clinical trial and is on the way to imminent availability, there are currently none offered for the treatment of approximately 6% of patients who may suffer a bleeding event. The work presented in the current manuscript is timely, clearly written and a well-rationalized biochemical tour-de-force that will be primarily of interest to other biochemists in the field, both from academia and the pharmaceutical industry. The strategy adopted by these authors, was based on a careful evaluation of the DOAC-FXa crystal structure. With additional insight gained from the amino acid sequence differences of snake venom FXa paralogs compared to human, the team deduced that a region within human FXa (the 91-99 loop) could be altered to selectively inhibit binding of DOACs to the active site of FXa and cause minimal effects on coagulation function. The 91-99 loop includes the known S4 substrate recognition Tyr99 residue. Altering this or inserting a small extended loop resulted in the predicted differential effects on DOAC versus coagulation function. The FXa variant that gave rise to the greatest inhibition of DOAC-binding with minimal alteration of FXa coagulation function was facilitated by inserting the analogous *Tropidechis carinatus* venom FXa sequence into the 91-99 loop (FXa-C). This is a completely novel approach resulting in a unique FXa construct.

Comments

1) While the biochemistry of the various FXa constructs generated by the authors were carefully evaluated from a biochemical perspective, it stops short of evaluation in an animal model. This could have easily propelled the work into the top five most significant papers published in the discipline this year. By altering the substrate specificity pocket of FXa it is possible that physiologically important proteins outside the normal substrate recognition of FXa may become new targets. Although not alluded to, evidence of new substrate targeting is suggested by the fragmentation profile presented for factor V (Fig. ED4). FXa-C treatment of factor V resulted in fragments at approximately 68 and 58 kDa, that were not visible in the wild type cleavage profile. Similarly, FXa-C did not produce a fragment typical of wild type FXa at ~49 kDa. An in

vivo model may demonstrate that potentially other new targets for Xa-C are irrelevant.

2) Fig. 1C shows the specific activity of FX using tissue factor-mediated initiation of coagulation (PT). Since it is known (Baroni, 2015 DOI:26012870) that distinct FX recognition sites exist for the analogous FX-activating complex facilitated by FVIIIa (APTT), this pathway of activation should also be reported for the FX variants.

3) Fig. 4B show differences in the proportion of FX α and FX β , and a lower molecular weight species, for wild type recombinant FX, FX-B and FX-C compared to plasma-derived FXa. Is this due to differences in the rate of activation by the RVV-X activator enabling differential autoproteolysis or to the susceptibility to autoproteolysis? Either explanation indicates alterations in FXa that confer enhanced cleavability. This could affect the interpretation of results and estimates of specific activity.

Minor Comments

- 1) What is the relative specific activity of plasma derived FX versus wild type recombinant FX?
- 2) It would be helpful in Fig. 1A(insert) to color differentiate Y99 and F174.
- 3) Tables should be reorganized so that the columns are consistent with their sequential mention in the text.

Reviewer #2 (Remarks to the Author):

In the present manuscript, Verhoef and coworkers have performed a combined experimental / computational study of Factor Xa (FXa), as well as FXa variants that have been engineered to retain their pro-coagulant activity. Overall, the paper is sound, and the topic is of moderate interest due to its biomedical applications. My main comments on the paper itself are minor; for example the last sentence of the second paragraph on pg. 8 is a bit awkward, it says that the *loop* is inhibited by apixaban. Or, for example, on p. 9, it is not clear what exactly the effect of the loop on inhibitor (apixaban) binding actually is, since even the authors say that the substrate displacement doesn't coincide with the loop movement, or vice versa. That is, they say that unfolding of the 99-loop coincides with little movement of apixaban, but then in the same paragraph say that steric hindrance imposed by the structurally flexible loop impairs apixaban binding. This calls for a more extended explanation. My biggest concern, however, is that while not scientifically unsound, despite the claims in the manuscript, the paper itself is incremental and mainly of interest to a very specialised audience, and the medical significance of the work is very preliminary. It would be perfectly suitable for a specialist biochemical journal, but I do not believe it has the scope and impact to be suitable for publication in Nature Communications, and as such cannot recommend it for publication.

Reviewer #3 (Remarks to the Author):

The work titled “Engineered Factor Xa Variants Retain Procoagulant Activity Independent of Direct Factor Xa Inhibitors” by Daniël Verhoef et al. represents a combined molecular dynamics and biochemical approach to evaluate the inhibition of blood coagulation factor Xa based on the engineered structural modifications. Based on in vitro biochemical work authors demonstrated that the amino acid substitutions: Y99A and F174A significantly increased the IC50 of FXa inhibition. The insertion of an additional loop segment, obtained from the partially inhibitor-resistant *P. textilis* FXa, between Y91 and D102 of human FXa, led to a reduced sensitivity toward FXa inhibitors, such as apixaban or rivaroxaban. Moreover, the loop modifications altered the catalytic activity of the FXa variants. The authors used a series of 100 ns MD simulations with CROMOS force field to provide structural rationale for the observed inhibitor resistance. This is clearly interesting work relating to the development of anticoagulant therapy. The rationale for this work and proposed experiments are technically sound and appropriate. However, there are number of issues which need to be addressed prior to consideration for publication.

1. The authors relied on MD simulations for the structural characterization of the reduced inhibition for Y99A and F174A variants of human XFa and also inhibition of *P. textiles* isoform FXa for which apixaban was docked “by hand”. The authors must provide docking details and validation of the isoform XFa – apixaban complex prior to drawing conclusions based on observed structural changes. The movement of the loop and/or inhibitor could be an artifact of docking. Also, the crystal structure (4BXW) used for the isoform simulation is complexed with coagulation factor V bound in proximity of the S4 subsite and PQR..DL loop region. The authors should consider such structural implications when conclusions are drawn for the human ortholog.
2. Based on calculated B-factors the authors reported high flexibility of the loop in the unbound isoform. These data should be included in Fig 3 as a control for the MD simulation of the complex. The conformational flexibility of the loop (in Fig3, A-D) can be reported by calculating RMSD (similar to apixaban) or by a separate plot with B values. This will give the reader a better understanding the extent of the motions relative to the unbound isoform.
3. All structural data supporting reduced inhibition are qualitative in nature. This work will greatly benefit from calculation of binding free energies (for example based on separate MD runs for ligand, complexes and receptor). It will provide quantitative description of binding and validate correlation between biochemical and structural data produced by MD, particularly for

Y99A and F174A single and double variants.

4. Since the Y99 and F174 are the part of the aromatic box with W215, which supports nonpolar interactions with aryl group of the inhibitor, this residue should be included in the structural analysis.

5. It is not clear why authors limit simulations to only 100 ns runs with only a single run for a wt FXa-inhibitor complex and unbound isoform FXa. The protein loop motions are often observed in the time scales up to 1 μ s. Longer simulations (0.5 to 1 μ s) could provide additional information and might be needed to properly equilibrate the system.

6. The authors should clarify which bonds were constrained with LINCS algorithm (X-H?) and explain their choice of the thermostat for temperature coupling.

7. It is not clear how the representative structures on fig.3 were generated? Are these time-averaged or selective snapshots? It should not be a last snapshot(s) from a 100 ns trajectory.

8. The Figure 1 needs clarification. What are the bars on Fig. 1A for interatomic distances? Are they standard deviations calculated from MD trajectories? Also, the std. deviations for IC50 are over 30% for variants and almost 100% for WT (Fig 1B). Do they represent only two repeats? Though standard deviation can be calculated based on two values, the use of more than 2 points is statistically more relevant for broad distribution. The same applies to other kinetic data in the paper with large std. dev.

Reviewer #1:

We thank the reviewer for his/her appreciation of the timeliness and novelty of our findings and for the helpful comments and expert input on our manuscript.

#1. While the biochemistry of the various FXa constructs generated by the authors were carefully evaluated from a biochemical perspective, it stops short of evaluation in an animal model. This could have easily propelled the work into the top five most significant papers published in the discipline this year. By altering the substrate specificity pocket of FXa it is possible that physiologically important proteins outside the normal substrate recognition of FXa may become new targets. Although not alluded to, evidence of new substrate targeting is suggested by the fragmentation profile presented for factor V (Fig. ED4). FXa-C treatment of factor V resulted in fragments at approximately 68 and 58 kDa, that were not visible in the wild type cleavage profile. Similarly, FXa-C did not produce a fragment typical of wild type FXa at ~49 kDa. An in vivo model may demonstrate that potentially other new targets for Xa-C are irrelevant.

We concur with the reviewer that the pattern of full-length plasma-derived factor V (pd-FV) fragmentation differs between wild-type FXa (wt-FXa) and FXa-C at supra-physiological concentrations of enzyme (≥ 10 nM). To explore whether the pd-FV cleavage pattern also differs at more physiologically relevant FXa concentrations, we have prolonged the incubations with 5 nM FXa to 60 minutes. The appearance of FVa heavy and light chain activation products over time was similar for both wt-FXa and FXa-C. Consistent with the previous findings, proteolysis of pd-FV by FXa-C appears to involve a cleavage site that is distinct from cleavage by wt-FXa, indicated by the presence of various partially activated FV fragments (FVa* in **Supplementary Fig. 7C,D**). We have now added these findings to our revised manuscript (“Proteolysis of full-length [...] (Supplementary Fig. 7).”, pg. 15) and in the revised Supplementary Information section (**Supplementary Fig. 7**, pgs. 9-10).

To address the reviewer’s concern regarding the possibility that FXa-C may target physiologically important proteins outside the normal FXa substrate recognition or may do so with enhanced efficiency, we have examined the FXa-dependent proteolysis of zymogens factor IX (**Fig. 1** of this rebuttal), XI (**Fig. 2** of this rebuttal), and protein C (**Fig. 3** of this rebuttal) under conditions

essentially similar to those of FV activation. No (enhanced) cleavage by FXa-C was observed, suggesting that the proteolytic activation of other new plasma proteins by FXa-C is irrelevant. This is in agreement with our observation that no significant surplus of thrombin was generated in the presence of supra-physiological concentrations of FX-C (**Fig. 5C,D** of the revised manuscript).

Fig. 1. Cleavage of factor IX by factor Xa variants. Plasma-derived factor IX (2.7 μ M; pd-FIX, Haematologic Technologies, Essex Junction, VT, USA) was incubated for 1 - 60 minutes with 5 nM recombinant human wt-FXa or FXa-C in the presence of 50 μ M PCPS at 25°C in assay buffer. Samples (3 μ g/lane) were subjected to SDS-PAGE under reducing conditions using the MES buffer system and visualized by staining with CBB. Lane 1: pd-FIX, no FXa; lanes 2-8: pd-FIX, incubated for 1, 2, 4, 10, 20, 40, and 60 minutes with wt-FXa or FXa-C. The protein bands corresponding to full-length pd-FIX, partially activated FIX (FIX α ; 45 kDa), the heavy chain of fully activated pd-FIX (FIXa-HC; 28 kDa), or the light chain of activated pd-FIX (FIXa-LC; 17 kDa) are indicated. The apparent molecular weights of the standards (kDa) are indicated. The data suggest that both FXa variants are capable of partially activating FIX into FIX α (\geq 40 minutes), and wt-FXa may be slightly more efficient in doing so. FIX α is known to display catalytic activity toward synthetic substrates only and does not have clotting activity.

Fig. 2. Cleavage of factor XI by factor Xa variants. Plasma-derived factor XI (0.9 μ M; pd-FXI, Haematologic Technologies, Essex Junction, VT, USA) was incubated for 1 - 60 minutes with 5 nM recombinant human wt-FXa or FXa-C in the presence of 50 μ M PCPS at 25°C in assay buffer. Samples (3 μ g/lane) were subjected to SDS-PAGE under reducing conditions using the MOPS buffer system and visualized by staining with CBB. Lane 1: pd-FXI, no FXa; lanes 2-8: pd-FXI, incubated for 1, 2, 4, 10, 20, 40, and 60 minutes with wt-FXa or FXa-C. The protein bands corresponding to full-length pd-FXI are indicated. The apparent molecular weights of the standards (kDa) are indicated. Conversion of the 80 kDa FXI zymogen subunit to the 50 and 30 kDa heavy and light chains of FXIa was not observed.

Fig. 3. Cleavage of protein C by factor Xa variants. Plasma-derived protein C (2.4 μ M; pd-PC, Haematologic Technologies, Essex Junction, VT, USA) was incubated for 1-60 minutes with 5 nM recombinant human wt-FXa or FXa-C in the presence of 50 μ M PCPS at 25°C in assay buffer. Samples (3 μ g/lane) were subjected to SDS-PAGE under reducing conditions using the MES buffer system and visualized by staining with CBB. Lane 1: pd-PC, no FXa; lanes 2-8: pd-PC, incubated for 1, 2, 4, 10, 20, 40, and 60 minutes with wt-FXa or FXa-C. The protein bands corresponding to the heavy chain of pd-PC (PC-HC; 41 kDa) and the light chain of pd-PC (PD-LC; 21 kDa) are indicated. The apparent molecular weights of the standards (kDa) are indicated. Conversion of the 41 kDa pd-PC heavy chain to the 35 kDa light chain of activated protein C was not observed. These findings were corroborated by assessment of peptidyl substrate S2366 (250 μ M final; Instrumentation Laboratories, Bedford, MA, USA) conversion (indicated as mOD/min) by time samples of pd-PC (66 nM final). No significant activation of pd-PC was observed given that the chromogenic activity in samples 2-8 corresponded to the conversion of S2366 by 5 nM of either wt-FXa or FXa-C.

We agree with the reviewer that in vivo validation would contribute significantly to the impact of our manuscript. However, preliminary evidence indicates that this evaluation necessitates the generation of species-specific FX variants to not only overcome the inherent differences between vertebrate coagulation systems, but also to correctly assess the efficacy and off-target effects of the FX molecules assessed. While we intend to pursue this approach, it is beyond the scope of the current manuscript.

#2. Fig. 1C shows the specific activity of FX using tissue factor-mediated initiation of coagulation (PT). Since it is known (Baroni, 2015 DOI:26012870) that distinct FX recognition sites exist for the analogous FX-activating complex facilitated by FVIIIa (APTT), this pathway of activation should also be reported for the FX variants.

We thank the reviewer for pointing this out and have now included the examination of the intrinsic activation pathway of the FX variants in reference to Baroni et al.

We investigated the activation of FX variants via the intrinsic (APTT) and extrinsic activation (PT) route in a new and independent experiment (**Fig. 1C**, revised manuscript pg. 32). In short, FX variants Y99A and Y99A/F174A displayed an impaired specific clotting activity compared to wt-FX, while this was enhanced for variant F174A. (“In addition, upon [...] Phe174Ala replacement (Fig. 1c).”, pg. 8, revised manuscript).

Activation of variant FX-C by the intrinsic FVIIIa/FIXa complex was also assessed, which has now been included as **Supplementary Fig. 6** in the revised Supplementary Information section (pg. 10) and is discussed in the revised manuscript (“Furthermore, the delay [...] were unperturbed (Supplementary Fig. 6).”, pgs. 15-16).

#3. Fig. 4B show differences in the proportion of FX α and FX β , and a lower molecular weight species, for wild type recombinant FX, FX-B and FX-C compared to plasma-derived FXa. Is this due to differences in the rate of activation by the RVV-X activator enabling differential autoproteolysis or to the susceptibility to autoproteolysis? Either explanation indicates alterations in FXa that confer enhanced cleavability. This could affect the interpretation of results and estimates of specific activity.

Initial test activations with RVV-X prior to batch activation and size-exclusion chromatography revealed no differences in the rate of and/or susceptibility for autoproteolysis of the FX variants relative to wt-FX. Evaluation of the batch activations made clear that the incubation times needed to be prolonged to accommodate full zymogen activation. Initially, wt-FX was activated for 60 minutes, which resulted in predominantly FX α prior to chromatography (Fig. 4A of this rebuttal). The incubation time was increased to 90 minutes for variant FXa-A, resulting in proportionally more FX β (Fig. 4B of this rebuttal). A prolonged incubation time of 120 minutes resulted in a 50/50 ratio of FX α /FX β for variants B and C (Fig. 4C,D of this rebuttal). Since the FX α /FX β ratio of the starting material was maintained for each variant during size-exclusion chromatography (Fig. 5 of this rebuttal), we assume that autoproteolysis and full conversion to FX β had occurred either during the final step of concentrating (to ~2mg/ml FX) and buffer exchange or during long term storage at -20°C. While the apparent rate of autoproteolysis was not different during initial RVV-X activation procedures, FXa-A did retain more FX α over time. This correlates with free FXa-A having the lowest catalytic efficiency towards both the peptidyl substrate and prothrombin as compared to variants B and C (Table 1, main manuscript). However, the altered FX α /FX β ratio has been acknowledged to not affect overall substrate recognition, as both isoforms are functionally similar with respect to prothrombinase assembly, prothrombin activation, antithrombin recognition, and peptidyl substrate conversion [Prydzial and Kessler, *J. Biol. Chem.* 1996, 271:16621-16626.]. We have provided this information in the caption of Fig. 4 (pg. 36) of our revised manuscript and thank the reviewer for bringing this issue to our attention.

Fig 4A. Size-exclusion chromatography and SDS-PAGE analysis of RVV-X-activated wild-type factor X. A sample of RVV-X-activated wt-FX was taken after batch activation, analyzed on reducing SDS-PAGE, and stained with CBB. The factor Xa α/β doublet is indicated by a bracket.

Fig. 4B. Size-exclusion chromatography and SDS-PAGE analysis of RVV-X-activated factor X variant A. A sample of RVV-X-activated FX-A was taken after batch activation, analyzed on reducing SDS-PAGE, and stained with CBB. The factor Xa α/β doublet is indicated by a bracket.

Fig. 4C. Size-exclusion chromatography and SDS-PAGE analysis of RVV-X-activated factor X variant B. A sample of RVV-X-activated FX-B was taken after batch activation, analyzed on reducing SDS-PAGE, and stained with CBB. The factor Xa α/β doublet is indicated by a bracket.

Fig. 4D. Size-exclusion chromatography SDS-PAGE analysis of RVV-X-activated factor X variant C. A sample of RVV-X-activated FX-C was taken after batch activation, analyzed on reducing SDS-PAGE, and stained with CBB. The factor Xa α/β doublet is indicated by a bracket.

Fig. 5. SDS-PAGE analysis of size-exclusion chromatography fractions. Fraction samples (40 μ l/lane) of the size-exclusion chromatography step of each FXa variant were subjected to SDS-PAGE under reducing conditions using the MES buffer system and visualized by staining with CBB. The protein bands corresponding to FXa α /FXa β appear as a doublet.

Minor Comment #1. *What is the relative specific activity of plasma derived FX versus wild type recombinant FX?*

We thank the reviewer for raising this issue as well. The relative specific extrinsic activity of purified wt-FX was similar to that of pd-FX (**Fig. 6A** of this rebuttal). The specific intrinsic activity of wt-FX, however, was significantly reduced relative to pd-FX (**Fig. 6B** of this rebuttal). This discrepancy in specific intrinsic activity is partly explained by the fact that pd-FX exhibited FXa activity upon incubation with Spectrozyme FXa, corresponding to 38 pM pd-FXa per 100 nM pd-FX. We speculate that these minute amounts of FXa would likely be able to activate FV during the longer incubation times of the APTT assay and thereby promote clotting, as opposed to the relative shorter incubation times in the PT assay. Indeed, addition of a similar trace of pd-FXa to wt-FX prior to APTT analysis significantly increased the specific APTT activity of wt-FX to 53% of pd-FX (**Fig. 6B** of this rebuttal; wt-FX*). No chromogenic FXa activity was observed in the presence of purified wt-FX. We would like to emphasize that in our assessment of the recombinant FX variants, wt-FX was always included as relevant control.

Fig. 6. Specific extrinsic and intrinsic clotting activity of purified factor X. The specific extrinsic (PT-based; panel A) or intrinsic (APTT-based; panel B) clotting activity of purified plasma-derived FX (pd-FX) or recombinant wild-type FX (wt-FX) is shown as the average \pm 1 standard deviation of three individual measurements per FX variant. * APTT clotting of wt-FX with 38 pM of pd-FXa / 100 nM of wt-FX added.

Minor Comment #2. *It would be helpful in Fig. 1A (insert) to color differentiate Y99 and F174.*

Separate color coding has now been applied for each of the three S4-subsite residues: Y99 (orange), F174 (cyan), and W215 (yellow) (**Fig. 1A**, revised manuscript pg. 32).

Minor Comment #3. *Tables should be reorganized so that the columns are consistent with their sequential mention in the text.*

The tables have been reorganized accordingly.

Reviewer #2:

We thank the reviewer for his/her careful review.

*#1. My main comments on the paper itself are minor; for example the last sentence of the second paragraph on pg. 8 is a bit awkward, it says that the *loop* is inhibited by apixaban.*

We would like to thank the reviewer for bringing this to our attention. We agree and used different language in the revised version of the manuscript (“In contrast, *P. textilis* [...] of apixaban (Fig. 2b).”, pg. 8).

#2. Or, for example, on p. 9, it is not clear what exactly the effect of the loop on inhibitor (apixaban) binding actually is, since even the authors say that the substrate displacement doesn't coincide with the loop movement, or vice versa. That is, they say that unfolding of the 99-loop coincides with little movement of apixaban, but then in the same paragraph say that steric hindrance imposed by the structurally flexible loop impairs apixaban binding. This calls for a more extended explanation.

This point and similar issues raised by reviewer #3 have been addressed by performing additional and substantially prolonged (750 ns) Molecular Dynamics (MD) simulations. These allowed us to extend the explanation on the steric hindrance between apixaban and the 99-loop, as described on pgs. 9-10 of the revised manuscript (“During these 750 ns ... ≥ 0.5 nm (Supplementary Fig. 3a-e).”, pg.9, and “In the other [...] impairing apixaban binding.”, pg. 10).

#3. My biggest concern, however, is that while not scientifically unsound, despite the claims in the manuscript, the paper itself is incremental and mainly of interest to a very specialised audience, and the medical significance of the work is very preliminary. It would be perfectly suitable for a specialist biochemical journal, but I do not believe it has the scope and impact to be suitable for publication in Nature Communications, and as such cannot recommend it for publication.

We respectfully disagree with the reviewer. Our findings not only demonstrate that lessons from nature are highly valuable, but also that they may form a framework for future medical applications. Here we have successfully pioneered the engineering of direct factor Xa inhibitor insensitivity in factor X(a) using a completely novel approach. The latter consisted of: i) careful evaluation of the inhibitor-FXa crystal structure employing Molecular Dynamics simulations, ii) in vitro validation by targeted mutagenesis in FX, and iii) combine this with naturally existing inhibitor-resistant snake venom FX species. As such, the generated inhibitor-resistant FX(a) protein could serve as a bypassing agent for factor Xa inhibitor-related bleeding. If proven efficacious, the medical urgency and impact of such a therapeutic protein will be significant. Therefore, this would certainly be of interest to the pharmaceutical industry.

As outlined in the Discussion section, the direct factor Xa inhibitors are widely prescribed, as millions of patients worldwide require anticoagulant drugs for the prophylactic management of stroke in atrial fibrillation or prevention and treatment of venous thrombosis. However, a major drawback to their use is the, thus far, absence of a worldwide-approved adequate reversal strategy to prevent and stop the associated potential life-threatening bleeding complications. Annually, 1-3% of the patients treated with anticoagulants suffer an adverse severe bleeding event, of which up to 1 in 5 are fatal. This underscores the urgent unmet clinical need for agents that counteract the direct inhibiting anticoagulants in case of bleeding or acute surgery.

Reviewer #3:

We would like to thank the reviewer for the comments provided, which, now addressed, have improved our manuscript. In order to address points 1-3 and 5, we re-simulated the systems for a total of 750 nanoseconds in several independent simulations starting from different random atomic velocities. The results and extended insights from these longer simulations are now included in the revised manuscript, as described below.

#1. The authors relied on MD simulations for the structural characterization of the reduced inhibition for Y99A and F174A variants of human FXa and also inhibition of P. textiles isoform FXa for which apixaban was docked “by hand”. The authors must provide docking details and validation of the isoform FXa – apixaban complex prior to drawing conclusions based on observed structural changes. The movement of the loop and/or inhibitor could be an artifact of docking. Also, the crystal structure (4BXW) used for the isoform simulation is complexed with coagulation factor V bound in proximity of the S4 subsite and PQQ...DL loop region. The authors should consider such structural implications when conclusions are drawn for the human ortholog.

We thank the reviewer for pointing to the omission of the docking details; the docking settings are now included in the revised manuscript (“Docking was performed [...] complex with apixaban.”, pg. 26). In addition, full details of the changes that were made to the Gln192 side chain conformation prior to docking have also been included (“To prevent sterical [...] in human FXa.” pg. 26). Validation of the docked ligand-binding pose is provided by the observed similarity between the docked pose and the binding pose of apixaban in the crystal structure of the human FXa-apixaban complex (“Initial docking of [...] human FXa (Supplementary Fig. 2).”, pg. 9 of the revised manuscript). Moreover, during our prolonged MD simulations we observed for one out of five replicas stable binding behavior of apixaban (**Fig. 3E**, revised manuscript). The observation that stable apixaban binding could be achieved during MD simulations further supports that the fast destabilization of binding (observed for four out of the five apixaban-bound isoform FXa MD simulations; **Fig. 3A-D**, revised manuscript) is not an artefact of the starting configuration, but results from the adopted 99-loop motions during MD simulation. In addition, we show that the observed 99-loop movements are similar for the inhibitor-bound and unbound MD simulations, further indicating that the loop-movement is not due to a docking artefact (**Supplementary Fig. 3**; “In the other [...] and unbound states.”, pg. 9 of the revised manuscript).

The MD simulations of isoform FXa are based on the isoform FXa crystal structure (PDB ID 4BXW) that was crystallized in the presence of a factor V peptide derived from the acidic C-terminal portion (a.a. 663-680) of the A2 domain of *P. textilis* coagulation factor V, described as the ‘fXa-a2 peptide complex’ by Lechtenberg et al. [Blood 2013, 122:2777-2783.]. Their X-ray data showed that this factor V peptide interacts with the positively charged heparin binding exosite in isoform FXa that is distant from both the S4 subsite and the PQQ...DL loop region, as indicated in **Fig. 7** of this rebuttal. This was confirmed by the co-crystal of isoform FXa and full-length *P. textilis* factor V (PDB ID 4BXS) that demonstrated a similar orientation of these critical regions (**Fig. 8**, this rebuttal). The fact that the binding site for the factor V peptide and our target regions, the S4 subsite and the PQQ...DL loop, are oriented at discrete protein regions in isoform FXa supports our use of the 4BXW crystal structure as model system.

Fig. 7. Crystal structure of isoform FXa in complex with the C-terminal A2-domain peptide of *P. textilis* factor V (PDB ID 4BXW). Indicated are the factor V peptide residues 672-680 (orange); FXa heparin binding exosite (blue), FXa catalytic triad (red), FXa S4 subsite residues (green), and FXa PQK...DL loop (yellow).

Fig. 8. Crystal structure of isoform FXa in complex with full-length *P. textilis* factor V (PDB ID 4BXS). Indicated are the factor V peptide residues 663-680 (orange); FXa heparin binding exosite (blue), FXa catalytic triad (red), FXa S4 subsite residues (green), and FXa PQK...DL loop (yellow).

While not relevant to the MD simulations, we have addressed the concern of the reviewer that full-length factor V may affect active site binding of apixaban in our human FXa variant FXa-C. Assaying the apixaban-inhibition of FXa-C in the presence and absence of the cofactor FVa demonstrated an almost 2-fold increase in the IC_{50} : from 840 ± 296 nM for apixaban inhibition of free FXa-C to 1535 ± 474 nM when assembled into prothrombinase. These findings indicate that prothrombinase-assembled FXa-C may be more resistant to inhibition by apixaban to some extent, which is likely due to structural constraints imposed by the interaction of FXa with the full-length cofactor FVa. We have now included these findings in the new **Supplementary Fig. 5** of the supplementary information section (see also “As the cofactor [...] FXa with FVa.”, pgs. 11-12 of

the revised manuscript). We think that inclusion of these data is informative and thank the reviewer for raising the issue.

#2. Based on calculated B-factors the authors reported high flexibility of the loop in the unbound isoform. These data should be included in Fig 3 as a control for the MD simulation of the complex. The conformational flexibility of the loop (in Fig3, A-D) can be reported by calculating RMSD (similar to apixaban) or by a separate plot with B values. This will give the reader a better understanding the extent of the motions relative to the unbound isoform.

We agree that inclusion of B-factors and atomic RMSDs of all bound and unbound simulations improves the understanding of the magnitude of the flexibility and motion of the extended 99-loop. We now included both RMSDs for the 99-loop and respective B-factors of all isoform simulations; the latter were plotted on their respective starting structures in the revised Supplementary Information section (**Supplementary Fig. 3**; see also pgs. 9-10 of the revised manuscript).

#3. All structural data supporting reduced inhibition are qualitative in nature. This work will greatly benefit from calculation of binding free energies (for example based on separate MD runs for ligand, complexes and receptor). It will provide quantitative description of binding and validate correlation between biochemical and structural data produced by MD, particularly for Y99A and F174A single and double variants.

While free-energy calculations would be of great additional value in the explanation of mutant effects, we would like to point out that free energy calculation would only be feasible by using free energy perturbation, thermodynamic integration, or other alchemical methods. Establishing such free energy calculations will be non-trivial, especially for a relatively flexible system like the FXa-apixaban binding interface and for large perturbations such as the change from aromatic side chain groups (in Y99/F174) to the methyl group in Alanine. Consequently, it will not be feasible to achieve converged results and a confidence interval within the range of observed experimental effects. We therefore opted to study the effects in a qualitative manner, and substantiated our findings by significantly increasing the number of simulations and prolonging their length. Note that the 99-loop extension system is particularly suitable for this purpose due to its stark effect on apixaban binding.

#4. Since the Y99 and F174 are the part of the aromatic box with W215, which supports nonpolar interactions with aryl group of the inhibitor, this residue should be included in the structural analysis.

We now have adjusted **Fig. 1A** and the structural analysis on pg. 7 of the revised manuscript to explicitly include W215.

#5. It is not clear why authors limit simulations to only 100 ns runs with only a single run for a wt FXa-inhibitor complex and unbound isoform FXa. The protein loop motions are often observed in the time scales up to 1 μ s. Longer simulations (0.5 to 1 μ s) could provide additional information and might be needed to properly equilibrate the system.

We agree with the reviewer that the possibility of extended loop motions on longer timescales are worth investigating. We now have included multiple simulations for the wt-FXa-inhibitor complex and re-simulated all systems during 750 ns on a GPU cluster using a faster version (5.1.4) of GROMACS. The results of the longer and additional simulations confirmed our previous conclusions on the role of the insertion helix in impairing inhibitor binding. The revised manuscript was adjusted accordingly (pgs. 9-10).

#6. The authors should clarify which bonds were constrained with LINCS algorithm (X-H?) and explain their choice of the thermostat for temperature coupling.

The Materials section has been edited to reflect the fact that all bond lengths were constrained during simulation, as is customary for simulations using the GROMOS force fields. Weak coupling methods (Berendsen coupling) were used to calibrate the GROMOS force fields. We therefore opted to use an improved version of weak coupling (*i.e.*, velocity-rescaling). See the ‘Molecular Dynamics simulations’ section (pgs. 26-27) of the revised manuscript.

#7. It is not clear how the representative structures on fig.3 were generated? Are these time-averaged or selective snapshots? It should not be a last snapshot(s) from a 100 ns trajectory.

The snapshots presented in **Fig. 3** are the conformations after 750 ns (originally 100 ns) of MD simulations. To provide further insight into the structure at earlier time points, we now have included the snapshots after 250 ns and 500 ns of simulation as well (**Supplementary Fig. 4**, revised Supplementary Information section).

#8. The Figure 1 needs clarification. What are the bars on Fig. 1A for interatomic distances? Are they standard deviations calculated from MD trajectories? Also, the std. deviations for IC50 are over 30% for variants and almost 100% for WT (Fig 1B). Do they represent only two repeats? Though standard deviation can be calculated based on two values, the use of more than 2 points is statistically more relevant for broad distribution. The same applies to other kinetic data in the paper with large std. dev.

We thank the reviewer for picking this up and would like to clarify. During the 750 ns MD simulations, minimal atomic (MI) distances, defined as the smallest atom-atom distance observed, were computed between the inhibitor apixaban and each of the residues in FXa. Each box-and-whisker plot shown in **Fig. 1A** displays the distribution of the MI distances in quartiles for the indicated FXa residue. To generate these quartiles, the dataset was sorted and partitioned in four groups of equal size. The inner wide box represents the inner two quartiles (or 50%) of the data, separated by the median that is indicated by the additional line. The bars represent the minimum and maximum value observed, and therefore implicitly indicate the regions of the other 50 % of the data. To clarify this, we have adapted the legend of **Fig. 1A** in the revised manuscript.

The std. deviations in **Fig. 1B** reflect the 95% confidence interval of the nonlinear regression fit to determine the IC₅₀. Wide confidence intervals were obtained for variants that had limited data-point coverage close to the IC₅₀ concentration, since the fit was calculated from 11 data points that were spread over a 10⁵ serial dilution range. We have used a similar inhibitor dilution range for each of the tested FXa variants to limit dilution bias.

We thank the reviewers again for their consideration and look forward to your decision on the manuscript.

Sincerely,

Mettine H.A. Bos
Assistant Professor
Leiden University Medical Center

Reviewers' Comments:

Reviewer #1 (Remarks to the Author):

The manuscript submitted by Dr. Bos and colleagues has been revised to address a number of the concerns that were noted in the first review. The manuscript is significantly improved and only the following very minor issue still remains:

1) While I agree that conducting animal experiments may be outside the scope of the current work, without these studies it is not accurate to state that “proteolytic activation of new plasma proteins by FXa-C is irrelevant”. The authors now allude to the difference between wtFX and Xa-C toward FV cleavage. This is an improvement over the first version that made no comment, but they still bury the novel sequence recognition by arguing that proper sized heavy and light subunits were nevertheless produced from FV. I appreciate the effort made by the authors to carefully eliminate several proteins as substrates (FIX, FXI and protein C), which should be mentioned in the revised manuscript. Although these data support the maintenance of selective FXa-C substrate specificity, the acquisition of additional protein targets cannot be excluded. This too should be mentioned in the revised manuscript.

Reviewer #3 (Remarks to the Author):

I am happy to see that the revised version of the manuscript accommodated most corrections and suggestions proposed by the reviewers. The use of the several 750 ns simulations clearly improved the MD part of the work. However, I disagree with the authors that it would be difficult to calculate binding free energies based on MD simulations ($dG = E_{\text{complex}} - E_{\text{ligand}} - E_{\text{receptor}}$ in which E is calculated based on individual simulations of complex, ligand and receptor). Since these energies are based on MM calculations, they are not absolute values and should not be used as such. However, this approach has been routinely used to obtain relative binding free energies for similar systems. It should characterize differences in apixaban binding to isoform FXa and human FXa. RMSD calculations are rather weak characterization of binding unless significant dissociation between ligand and receptor is observed. Structural coupling between inhibitor and protein could be evaluated by the Pearson correlation coefficient rather than just RMSD values. However, the conclusion made in this work that the specific modifications of the substrate binding pocket of the FXa active site disrupt high affinity of FXa inhibitors is well supported by biochemical data. The lack of the binding energies is not crucial in this case.

Referee Response by Verhoef *et al.* for NCOMMS-17-00117A

Reviewer #1:

The manuscript submitted by Dr. Bos and colleagues has been revised to address a number of the concerns that were noted in the first review. The manuscript is significantly improved and only the following very minor issue still remains:

While I agree that conducting animal experiments may be outside the scope of the current work, without these studies it is not accurate to state that “proteolytic activation of new plasma proteins by FXa-C is irrelevant”. The authors now allude to the difference between wtFX and Xa-C toward FV cleavage. This is an improvement over the first version that made no comment, but they still bury the novel sequence recognition by arguing that proper sized heavy and light subunits were nevertheless produced from FV. I appreciate the effort made by the authors to carefully eliminate several proteins as substrates (FIX, FXI and protein C), which should be mentioned in the revised manuscript. Although these data support the maintenance of selective FXa-C substrate specificity, the acquisition of additional protein targets cannot be excluded. This too should be mentioned in the revised manuscript.

Response: We thank the reviewer for pointing out this omission in our manuscript, which we have now adapted according to the reviewer’s suggestions.

The text on page 15 of the revised manuscript is more specific in pointing out that there are differences in the fragmentation patterns of FV cleavage between wt-FXa and FXa-C, which is supported by marking of these specific fragments (~48-68 kDa) in Supplementary Fig. 6. In addition, we have deleted ‘nevertheless’ and included a statement on the acquisition of additional protein targets. The cleavage analysis of factors IX, XI, and protein C has now been included in the Supplementary Information. The adapted section on pg. 15-16 is as follows:

“While proteolysis of full-length plasma-derived FV by FXa-C did result in generation of the FVa heavy chain and light chain activation products, FXa-C appeared to cleave FV at one or more sites that are distinct from cleavage by wt-FXa, indicated by a differential fragmentation profile (Supplementary Fig. 6). The altered substrate recognition of FXa-C was further assessed by cleavage analyses of factors IX, XI, and protein C (Supplementary Figs. 7-9), which suggested that FXa-C maintains selective substrate specificity. However, the potential acquisition of additional protein targets cannot be excluded.”

Reviewer #3:

I am happy to see that the revised version of the manuscript accommodated most corrections and suggestions proposed by the reviewers. The use of the several 750 ns simulations clearly improved the MD part of the work. However, I disagree with the authors that it would be difficult to calculate binding free energies based on MD simulations ($dG = E_{\text{complex}} - E_{\text{ligand}} - E_{\text{receptor}}$ in which E is calculated based on individual simulations of complex, ligand and receptor). Since these energies are based on MM calculations, they are not absolute values and should not be used as such. However, this approach has been routinely used to obtain relative binding free energies for similar systems. It should characterize differences in apixaban binding to isoform FXa and human FXa. RMSD calculations are rather weak characterization of binding unless significant dissociation between ligand and receptor is observed. Structural coupling between inhibitor and protein could be evaluated by the Pearson correlation coefficient rather than just RMSD values. However, the conclusion made in this work that the specific modifications of the substrate binding

pocket of the FXa active site disrupt high affinity of FXa inhibitors is well supported by biochemical data. The lack of the binding energies is not crucial in this case.

Response: We agree with the reviewer that the prolonged MD simulations have improved our manuscript, and we think that the multiple long MD simulations together with the presented analyses give useful insights into the effects of the presented mutations on apixaban binding. We still do not believe in the possibility to obtain a reliable ΔG estimate from differences in energy terms (as suggested by the reviewer), because changes in entropy can be expected to also play a role for this flexible system and there may well be issues in obtaining converged values for the various energy terms. We agree with the reviewer that computing differences in binding free energies is not crucial here.

We thank the reviewers again for their consideration.

Sincerely,

Mettine H.A. Bos
Assistant Professor
Leiden University Medical Center